# T cell-specific deletion of *Pgam1* reveals a critical role for glycolysis in T cell responses

Koji Toriyama[1,2], Makoto Kuwahara[2], Hiroshi Kondoh[3], Takumi Mikawa[3], Nobuaki Takemori[4], Amane Konishi[2,5], Toshihiro Yorozuya[5], Takeshi Yamada[6,7], Tomoyoshi Soga[8], Atsushi Shiraishi[1] & Masakatsu Yamashita [2,6,9]✉

Although the important roles of glycolysis in T cells have been demonstrated, the regulatory mechanism of glycolysis in activated T cells has not been fully elucidated. Furthermore, the influences of glycolytic failure on the T cell-dependent immune response in vivo remain unclear. We therefore assessed the role of glycolysis in the T cell-dependent immune response using T cell-specific *Pgam1*-deficient mice. Both CD8 and CD4 T cell-dependent immune responses were attenuated by Pgam1 deficiency. The helper T cell-dependent inflammation was ameliorated in *Pgam1*-deficient mice. Glycolysis augments the activation of mTOR complex 1 (mTORC1) and the T-cell receptor (TCR) signals. Glutamine acts as a metabolic hub in activated T cells, since the TCR-dependent increase in intracellular glutamine is required to augment glycolysis, increase mTORC1 activity and augment TCR signals. These findings suggest that mTORC1, glycolysis and glutamine affect each other and cooperate to induce T cell proliferation and differentiation.

[1] Department of Ophthalmology, Graduate School of Medicine, Ehime University, Shitsukawa, Toon, Ehime 791-0295, Japan. [2] Department of Immunology, Graduate School of Medicine, Ehime University, Shitsukawa, Toon City, Ehime 791-0295, Japan. [3] Geriatric Unit, Graduate School of Medicine, Kyoto University, Sakyo-ku, Kyoto 606-8507, Japan. [4] Advanced Research Center, Ehime University, Shitsukawa, Toon City, Ehime 791-0295, Japan. [5] Department of Anesthesia and Perioperative Medicine, Graduate School of Medicine, Ehime University, Shitsukawa, Toon City, Ehime 791-0295, Japan. [6] Department of Infection and Host Defenses, Graduate School of Medicine, Ehime University, Shitsukawa, Toon City, Ehime 791-0295, Japan. [7] Department of Medical Technology, Ehime Prefectural University of Health Sciences, Tobe City, Ehime 791-0295, Japan. [8] Institute for Advanced Bioscience, Keio University, Tsuruoka City, Yamagata 997-0052, Japan. [9] Department of Translational Immunology, Translational Research Center, Ehime University Hospital, Shitsukawa, Toon City, Ehime 791-0295, Japan. ✉email: yamamasa@m.ehime-u.ac.jp

On antigen recognition, naïve T cells initiate a cell-intrinsic program that induces a proliferative response and differentiation into effector T cells. Recent evidence suggests that T cells dramatically alter their metabolic activity during T cell receptor (TCR)-mediated activation[1–3]. Non-proliferating naïve T cells use fatty acid oxidization and/or a low rate of glycolysis and subsequently oxidize glucose-derived pyruvate via oxidative phosphorylation (OXOPHOS) to generate ATP[4,5]. In contrast, the activated T cells rely on an increasing rate of glycolysis despite the aerobic condition which is required for rapid cellular proliferation and acquisition of the effector function[4,6], known as the Warburg effect[7]. Although effector T cells dominantly use aerobic glycolysis for glucose metabolism, OXPHOS also continues to occur[6,8]. This change in the metabolic status is termed "metabolic reprogramming" and is believed to play an important role in the regulation of T cell-mediated immune responses[9].

Phosphoglycerate mutase 1 (Pgam1) is a glycolytic enzyme that catalyzes the conversion of 3-phosphoglycerate (3-PG) to 2-phosphoglycerate (2-PG). Pgam1 is thought to regulate both glycolysis and biosynthesis, since 3-PG inhibits 6-phosphogluconate dehydrogenase in the pentose phosphate pathway (PPP) while 2-PG activates 3-phosphoglycerate dehydrogenase in serine synthesis[10]. Recent studies have revealed that Pgam1 is upregulated in several human cancers and linked to the tumor growth and survival[11,12]. As described above, the metabolic profiles of activated T cells are similar to the profile in tumors, represented by aerobic glycolysis. Hence, Pgam1 also plays an important role in regulating the metabolic reprogramming of T cells.

The mechanistic target of rapamycin (mTOR) is a crucial regulator of cellular metabolism[13,14]. mTOR signaling is essential for immune signals and alteration of cellular metabolism for proper differentiation and function of T cells[15–17]. mTOR exists in two complexes, mTOR complex 1 (mTORC1) and mTORC2. mTORC1 and mTORC2 orchestrate metabolic reprograming to exit the quiescent state of T cells[18]. mTORC1 regulates protein synthesis via phosphorylation of the S6 kinases and the inhibitory eukaryotic initiation factor 4E (eIF4E)-binding proteins. It was reported that the mTORC1-dependent induction of Myc promotes glucose and glutamine metabolism in activated T cells[8,19]. Myc-dependent metabolic reprograming promotes the T-cell proliferative response and differentiation[18,20]. Thus, a consequence of increased mTORC1 activity during quiescence exit is the increased translation of pro-anabolic enzymes and transcription factors, such as Myc[21].

Proliferating cells vigorously import extracellular glutamine which supply carbon and nitrogen for the biomass accumulation[22,23]. T cells also increase glutamine take up after TCR-stimulation, and glutamine promote T-cell activation and proliferation[8,24]. In the glutaminolysis, glutamine is converted to glutamate, and then, glutamate is catabolized to α-ketoglutarate (α-KG), which is consumed through tricarboxylic acid (TCA) cycle[25,26]. Glutamine is thought to activate T cells through several mechanisms. Recently, we reported that glutamine activates mTORC1 signaling, partly via supplementation of α-KG[27].

In the present study, we examined the role of glycolysis using T-cell-specific Pgam1 knockout (KO) mice (Pgam1[flox/flox] with CD4-Cre transgenic) and found that glycolysis, mTORC1 and glutamine affect each other and cooperate to induce T-cell proliferation and differentiation.

## Results

**Glycolysis controls TCR-mediated signal transduction**. Upon antigen recognition, T cells show a dramatic increase in glucose metabolism[1–3]. However, the influence of glycolytic failure on the T-cell-dependent immune response in vivo is poorly understood.

The mRNA expression of pgam1 was induced by TCR-stimulation in CD8 T cells, whereas the level of pgam2 mRNA, an isozyme of Pgam1, was decreased (Supplementary Fig. 1a). We therefore generated T cell-specific Pgam1 KO mice to clarify the roles of glycolysis during TCR-mediated activation and the T-cell-dependent immune response. The reduction in Pgam1 protein in Pgam1 KO CD8 T cells was confirmed by immunoblotting (Supplementary Fig. 1b). Pgam1 deficiency showed no effect on thymic T cell development or the T cell number in the spleen (Supplementary Fig. 2a). The memory/activated phenotype CD4 and CD8 T cells were marginally decreased in Pgam1 KO mice compared with wild-type mice (Supplementary Fig. 2b). The numbers of Foxp-positive CD4 T cells and invariant NKT cells were decreased in the spleen, whereas the numbers of these cells in the thymus and mesenteric lymph node were comparable (Supplementary Fig. 2c, d). Interleukin (IL)-2 production was significantly lower in Pgam1 KO CD4 T cells than in wild-type CD4 T cells. (Supplementary Fig. 2e).

We first assessed the metabolic profile in Pgam1 KO T cells using an extracellular flux analyzer. The glycolysis assessed by the extracellular acidification rate (ECAR) at 24 h after TCR-stimulation was lower in Pgam1 KO activated CD8 T cells than in wild-type cells (Fig. 1a). Pgam1 KO activated CD4 T cells also showed reduced ECAR compared with wild-type (Supplementary Fig. 3a). The basal oxygen consumption rate (OCR) and spare respiratory capacity (SRC) at 24 h after TCR-stimulation showed a significant reduction under conditions of Pgam1 deficiency in CD8 T cells (Fig. 1b). The basal OCR in Pgam1 KO activated CD4 T cells was comparable to that in wild-type CD4 T cells, whereas the SRC was decreased in Pgam1 KO CD4 T cells (Supplementary Fig. 3b). The ECAR in Pgam1 KO activated CD8 T cells at 8 h was comparable to that in wild-type CD8 T cells (Supplementary Fig. 4a). In addition, the basal OCR at 8 h in Pgam1 KO activated CD8 T cells was comparable to that in wild-type CD8 T cells, whereas the SRC was decreased in Pgam1 KO CD8 T cells (Supplementary Fig. 4b). The intracellular concentration of glycolytic intermediates before the Pgam-dependent catabolizing step (G6P, F6P, F1-6P DHAP, and 3PG) at 24 h after TCR-mediated activation was increased in Pgam1 KO CD8 T cells in comparison to wild-type CD8 T cells (Fig. 1c). In contrast, the intracellular level of lactate, an end product of anaerobic glycolysis, was decreased in Pgam1 KO cells (Fig. 1c). Pgam1-deficiency only showed a marginal effect on the intracellular concentrations of glycolytic products at 6 h after stimulation (Fig. 1c). These results were consistent with the expression pattern of pgam1 mRNAs that demonstrated the shift from pgam2 to pgam1 upon the TCR-mediated activation of CD8 T cells (Supplementary Fig. 1a). The concentrations of TCA cycle intermediates, succinate, fumarate, and malate were decreased in Pgam1-deficient CD8 T cells at 24 h after stimulation, whereas the levels of citrate and cis-aconitate were not affected by Pgam1 deficiency (Supplementary Fig. 4c). The intracellular concentrations of TCA cycle intermediates at 6 h were moderately increased in Pgam1 KO CD8 T cells in comparison to wild-type CD8 T cells (Supplementary Fig. 4c). The intracellular amounts of both NAD+ and NADH at 24 h were significantly decreased in Pgam1 KO CD8 T cells in comparison to wild-type cells, although these concentrations were comparable at 6 h (Supplementary Fig. 4d). The intracellular concentration of intermediates of the pentose phosphate pathway (PPP) at 24 h was moderately increased in Pgam1 KO CD8 T cells in comparison to wild-type cells (Supplementary Fig. 4e), and the intracellular levels of IMP, AMP, GMP, and UMP were reduced by Pgam1 deficiency (Supplementary Fig. 4f). These results suggest that nucleotide synthesis, but not PPP, is inhibited by Pgam1 deficiency in activated CD8 T cells. The intracellular concentration of ATP in Pgam1 KO CD8 T cells was equivalent to that in wild-type CD8

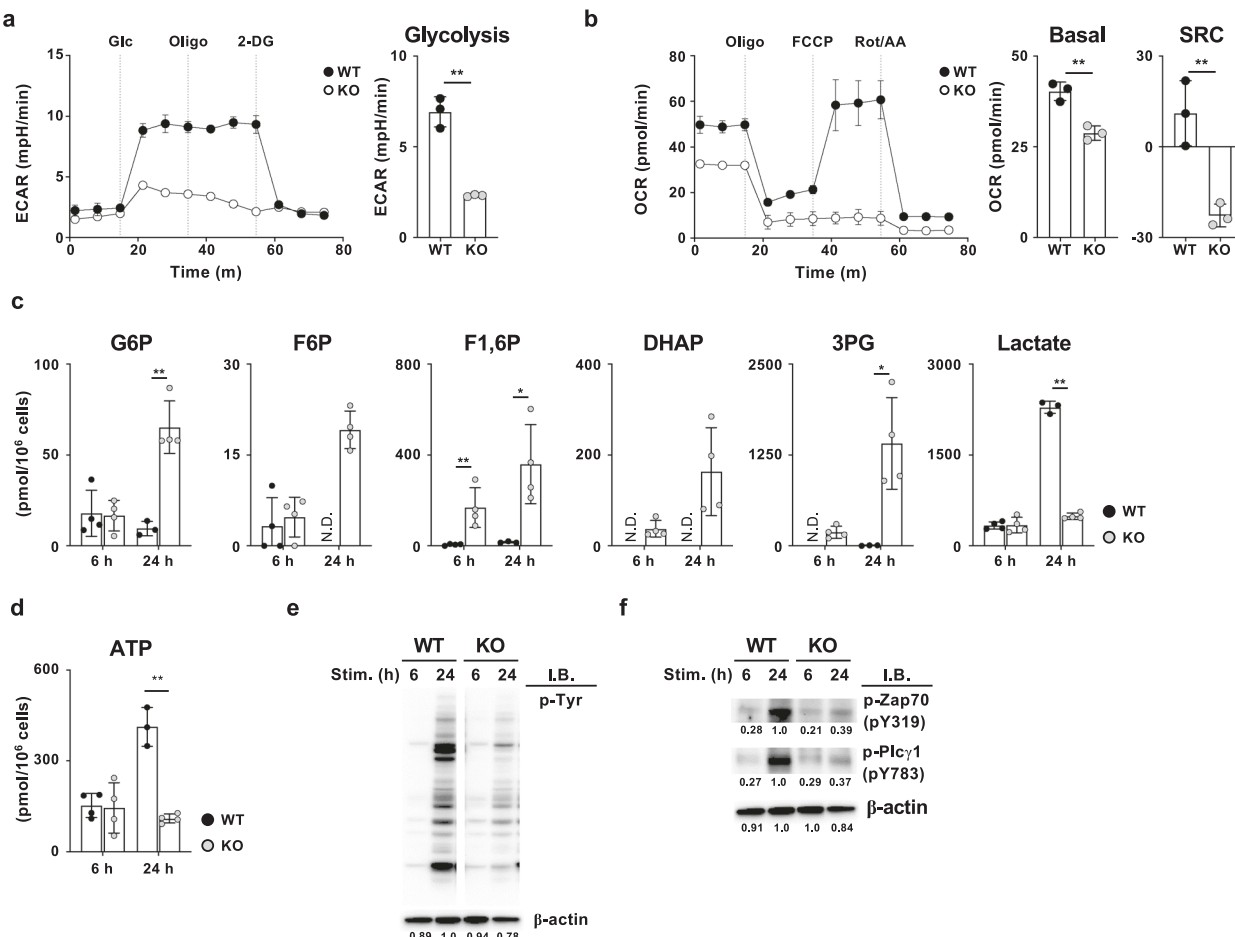

**Fig. 1 Glycolysis controls TCR-mediated signal transduction. a, b** WT and *Pgam1* KO naive CD8 T cells were stimulated for 24 h, and then **a** the extracellular acidification rate (ECAR) and **b** oxygen consumption rate (OCR) were determined ($n = 3$, technical replicate). Error bars represent the standard deviation. The results of the analyses are representative of at least three independent experiments with similar results. **c** WT and *Pgam1* KO naïve CD8 T cells were stimulated with anti-TCR-β and anti-CD28 mAbs for 6 or 24 h, and the intracellular amounts of glycolytic intermediates were determined by metabolic profiling ($n = 3$–4, biological replicates). **d** The intracellular amount of ATP of the cells in **c**. **e** The phospho-tyrosine and β-actin levels in WT and *Pgam1* KO CD8 T cells stimulated for indicated periods were determined by an immunoblot analysis. The protein amount of β-actin was used as a loading control. The numbers below the bands indicate the densitometry ratio with WT stimulated for 24 h. **f** The results of the immunoblot analysis of phospho-Zap70 (Tyr319), phosphor-Plcγ1 (Tyr783), and β-actin (control) in cells in **e** are shown. The numbers below the bands indicate the densitometry ratio with WT stimulated for 24 h. The results of the immunoblot analyses are representative of at least three independent experiments with similar results and are presented as cropped images. The full-length blots are presented in Supplementary Fig. 18. *$P < 0.05$, **$P < 0.01$ (Student's *t* test).

T cells at 6 h after TCR stimulation (Fig. 1d). While the level of ATP was further increased in wild-type CD8 T cells at 24 h, it was not increased but instead decreased in *Pgam1* KO CD8 T cells (Fig. 1d).

ATP is well known to act as a phosphate donor and is required for the protein kinase activity[28]. We therefore assessed the protein-tyrosine phosphorylation status of *Pgam1* KO activated CD8 T cells. The protein tyrosine phosphorylation level was lower in *Pgam1* KO activated CD8 T cells than in wild-type cells at 24 h after TCR stimulation, whereas the level at 6 h was comparable in both cell types (Fig. 1e). The tyrosine phosphorylation status of the TCR signaling molecules Zap70 (Tyr319) and Plcγ1 (Tyr783) was also decreased in *Pgam1* KO activated CD8 T cells at 24 h after stimulation and the level at 6 h was equivalent to that in wild-type cells. (Fig. 1f). These results indicate that the glycolytic activity and the intracellular concentration of ATP are correlated with the signal strength in activated CD8 T cells.

**Glycolysis is required for the development of effector CD8 T cells.** Wild-type and *Pgam1* KO CD8 T cells were stimulated

with anti-TCRβ mAb plus anti-CD28 mAb for 36 h, and the expression of activation makers was assessed by FACS. The expression of CD25, IL-2 receptor α chain, was decreased in *Pgam1* KO CD8 T cells compared to wild-type CD8 T cells, whereas the CD44 and CD69 levels were unaffected by Pgam1 deficiency (Fig. 2a). Consistent with the decreased expression of CD25, the IL-2-dependent phosphorylation of Stat5 was reduced in *Pgam1* KO CD8 T cells in comparison to the wild-type CD8 T cells (Fig. 2b), indicating that the reactivity of activated CD8 T cells to IL-2 was dependent on glycolytic activation. The TCR-dependent proliferative response was decreased in *Pgam1* KO CD8 T cells compared to wild-type CD8 T cells in vitro (Fig. 2c). Furthermore, *Pgam1* KO mice showed a lower expansion of OVA-specific CD8 T cells in the spleen after OVA-peptide-expressing *Listeria monocytogenes* (*Lm*-OVA) infection (Fig. 2d and Supplementary Fig. 5). Thus, the T-cell expansion was attenuated by Pgam1 deficiency both in vitro and in vivo. The essential role of de novo nucleotide synthesis in the cell cycle progression of activated T cells has been reported[29]. We demonstrated the decrease in the intracellular nucleotides

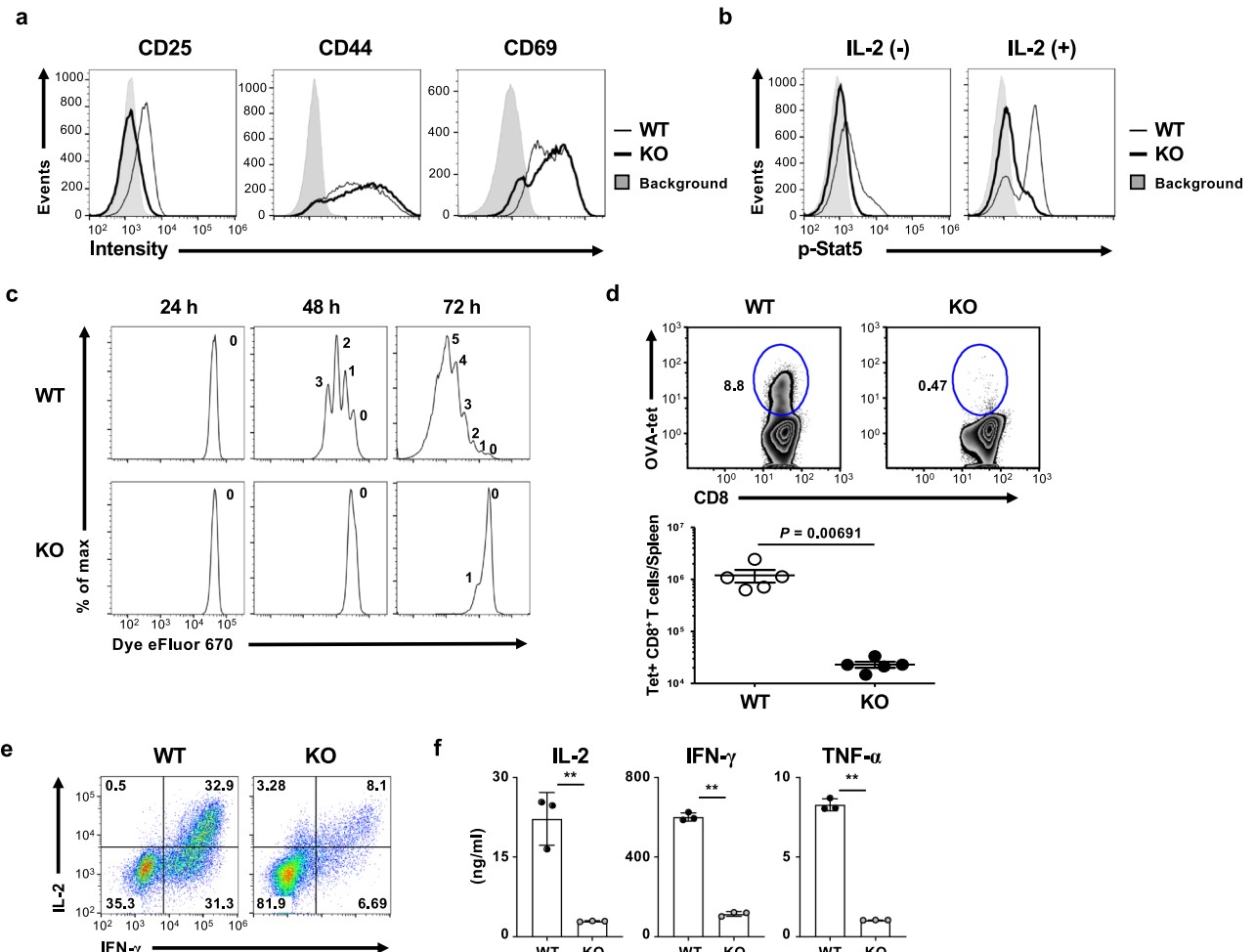

**Fig. 2 Glycolysis is required for the development of effector CD8 T cells. a** Representative staining profile of CD25, CD44, and CD69 in WT and *Pgam1* KO CD8 T cells stimulated with anti-TCR-β plus anti-CD28 mAbs for 36 h. **b** Representative staining profiles of phospho-STAT5 (Tyr694) in activated CD8 T cells. WT and *Pgam1* KO naïve CD8 T cells were stimulated with anti-TCR-β mAb plus anti-CD28 mAb for 48 h, and then the cells were cultured with or without IL-2 for 24 h. **c** WT and *Pgam1* KO naive CD8 T cells were labeled with eFluor670 and stimulated with anti-TCR-β mAb plus anti-CD28 mAb. The cell division was detected by flow cytometry at the indicated number of hours after the initial stimulation. **d** The immune response of antigen-specific CD8 T cells after *Lm*-OVA infection was analyzed by staining with an OVA-specific tetramer (Tet). A representative staining profile of Tet/CD8 gated on the CD8-positive cells in the spleen at 7 days post infection (upper). The percentages of cells are indicated in the circle. The absolute number of Tet$^+$ CD8 T cells in the spleen is shown (lower). Each point represents an individual mouse. **e** Representative results of the intracellular FACS analysis of IFN-γ/IL-2 in the WT and *Pgam1* KO CD8 T cells cultured under IL-2 conditions on day 5. The percentages of cells are indicated in each quadrant. **f** The results of an ELISA for IL-2, IFN-γ, and TNF-α in the supernatants of the cells in **e** ($n = 3$, biological replicate). The results of the analyses are representative of at least three independent experiments with similar results. The results are indicated with the standard deviation. *$P < 0.05$, **$P < 0.01$ (Student's *t*-test).

concentration in *Pgam1* KO activated CD8 T cells (Supplementary Fig. 4e), which would result in an impaired T-cell proliferative response.

We next assessed the effector differentiation in *Pgam1* KO CD8 T cells in vitro. The production of IL-2, interferon-γ (IFN-γ) and tumor necrosis factor-α (TNF-α) in *Pgam1* KO CD8 T cells was significantly lower than in wild-type CD8 T cells (Fig. 2e, f). The impairment of effector differentiation in *Pgam1* KO CD8 T cells was not restored by the supplementation of pyruvate, the end product of glycolysis (Supplementary Fig. 6a, b). These results suggest that glycolysis is required for maximal proliferation and effector differentiation of CD8 T cells.

**Impaired TH2 and TH17 cell differentiation in Pgam1 KO naïve CD4 T cells.** We next focused on the role of glycolysis in effector CD4 T cell development. In contrast to *Pgam1*-deficient CD8 T cells, *Pgam1* KO CD4 T cells showed only a moderate

reduction in CD25 (Supplementary Fig. 7a). The expression of CD44 and CD69 in *Pgam1* KO CD4 T cells was comparable to that in wild-type CD4 T cells, whereas the expression of CD40L showed a marked reduction (Supplementary Fig. 7a). The TCR-dependent proliferative response was decreased in *Pgam1* KO CD4 T cells compared to wild-type CD4 T cells in vitro (Supplementary Fig. 7b). Consistent with the decreased expression of CD40L in *Pgam1* KO activated CD4 T cells, the serum levels of dinitrophenyl (DNP)-specific IgM, IgG1, and IgG2c but not IgA in 2,4-dinitrophenylated ovalbumin (DNP-OVA)-immunized *Pgam1* KO mice were lower than in wild-type mice (Supplementary Fig. 8).

We next assessed the differentiation to effector helper T (Th) cell subsets in *Pgam1* KO naïve CD4 T cells in vitro. The generation of IFNγ-producing T$_H$1 cells was moderately decreased in the *Pgam1* KO naïve CD4 T cells compared to the wild-type naïve CD4 T cells (Fig. 3a). The production of IFN-γ and TNF-α in in vitro differentiated *Pgam1* KO T$_H$1 cells was

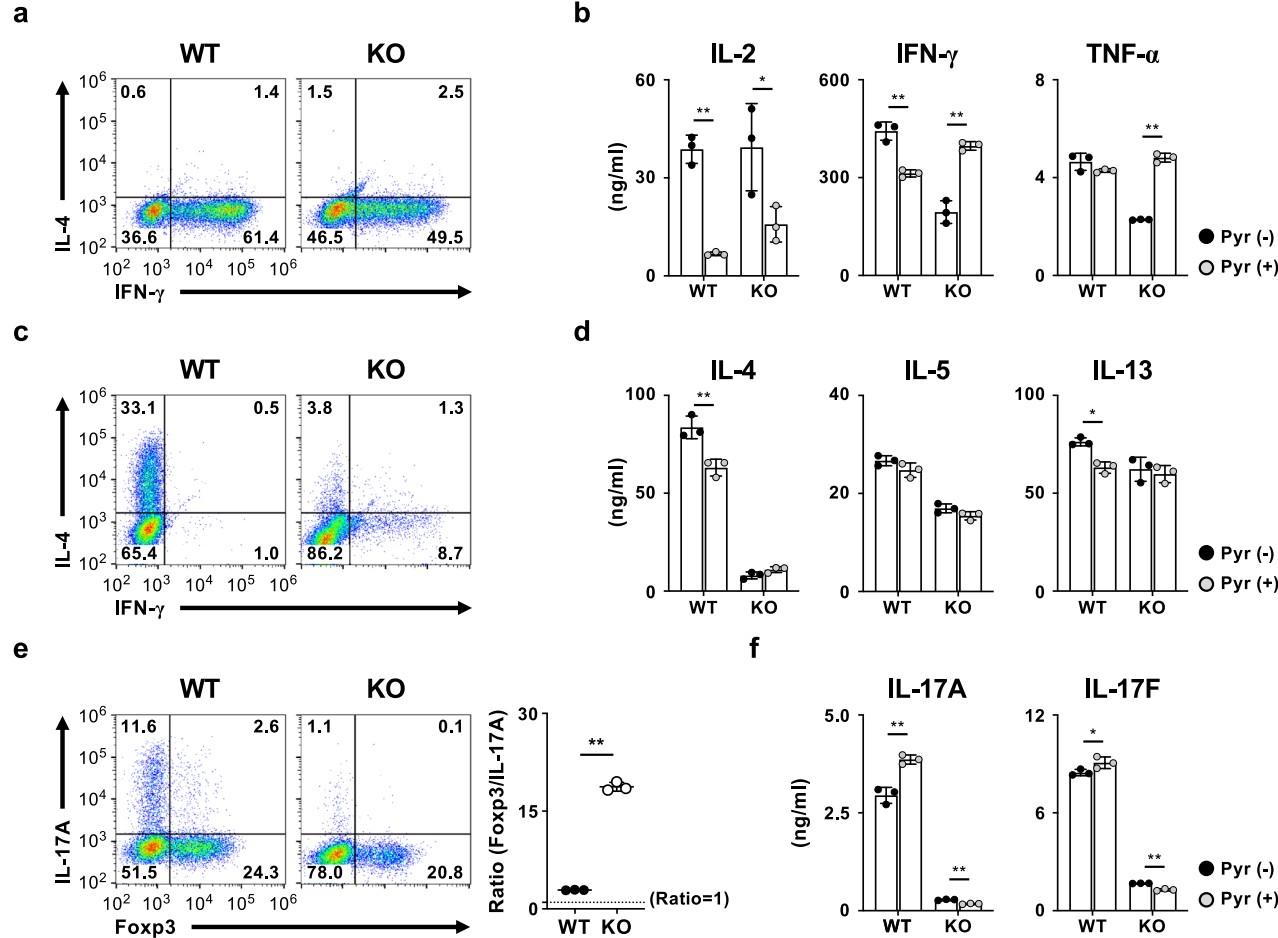

**Fig. 3 Impaired TH2 and TH17 cell differentiation in Pgam1 KO naïve CD4 T cells. a** Representative results of the intracellular FACS analysis of IFN-γ/IL-4 in the WT and *Pgam1* KO CD4 T cells cultured under Th1 conditions for 5 days. **b** WT and *Pgam1* KO CD4 T cells cultured under Th1 conditions in the presence or absence of pyruvate for 5 days. Then the cells were restimulated with immobilized ani-TCRβ mAb for 16 h. The amounts of IL-2, IFN-γ, and TNF-α in the supernatants was determined by an ELISA ($n = 3$, biological replicate). **c** Representative results of the intracellular FACS analysis of IFN-γ/IL-4 in the WT and *Pgam1* KO CD4 T cells cultured under Th2 conditions for 5 days. **d** WT and *Pgam1* KO CD4 T cells cultured under Th2 conditions in the presence or absence of pyruvate for 5 days. Then the cells were restimulated with immobilized ani-TCRβ mAb for 16 h. The amounts of IL-4, IL-5, and IL-13 in the supernatants was determined by an ELISA ($n = 3$, biological replicate). **e** Representative results of the intracellular FACS analysis of Foxp3/IL-17A in the WT and *Pgam1* KO CD4 T cells cultured under Th17 condition for 3 days (left panel). The ratio of Foxp3-positive cells/IL-17A-positive cells is shown in the right panel. **f** WT and *Pgam1* KO CD4 T cells cultured under Th17 conditions in the presence or absence of pyruvate for 3 days. Then the cells were restimulated with immobilized ani-TCRβ mAb for 16 h. The amounts of IL-17A and IL-17F in the supernatants was determined by an ELISA ($n = 3$, biological replicate). The results of the FACS analyses are representative of at least three independent experiments with similar results. The percentages of cells are indicated in each quadrant. The results are indicated with the standard deviation. *$P < 0.05$, **$P < 0.01$ (Student's $t$-test).

decreased to half, whereas the production of IL-2 was equivalent to that in wild-type $T_H1$ cells (Fig. 3b). The generation of IL-4-producing $T_H2$ cells was severely decreased, and the induction of IFN-γ-producing cells was detected in *Pgam1* KO naïve CD4 T cells cultured under $T_H2$ conditions (Fig. 3c). Although the decreased IL-4 production was confirmed by an enzyme-linked immunosorbent assay (ELISA), the production of IL-5 and IL-13 showed only a moderate reduction in *Pgam1* KO $T_H2$ cells (Fig. 3d). *Pgam1* KO naïve CD4 T cells cultured under $T_H17$ conditions showed a striking reduction in the generation of IL-17A-producing cells compared with wild-type naive CD4 T cells (Fig. 3e). The generation of Foxp3-positive regulatory T (Treg) cells was slightly decreased, so the ratio of Foxp3-postive cells and IL-17A-producing cells was increased (Fig. 3e). The decreased production of IL-17A and IL-17F was confirmed by an ELISA (Fig. 3f). Although the production of $T_H1$ cytokine (IFN-γ and TNF-α) in *Pgam1* KO CD4 T cells was restored by the

supplementation of pyruvate (Fig. 3b), $T_H2$ and $T_H17$ cell differentiation in *Pgam1* KO CD4 T cells was insensitive to pyruvate supplementation (Fig. 3d, f). Interestingly, the production of IL-2 was suppressed by pyruvate supplementation in both wild-type and *Pgam1* KO Th1 cells (Fig. 3b).

**Attenuated helper T-cell-dependent inflammation in Pgam1 KO mice.** Next, we assessed $T_H$ cell-dependent inflammatory response in *Pgam1* KO mice. To assess the impact of glycolytic inhibition on the in vivo $T_H2$ immune-response, allergic airway inflammation was induced in wild-type and *Pgam1* KO mice by the intranasal administration of OVA following immunization with OVA. A significant decrease in mononuclear cells infiltrating the peribronchiolar regions of the lungs was observed in the *Pgam1* KO mice (Fig. 4a). The bronchioles of the *Pgam1*-deficient mice showed less mucus production and goblet cell neoplasia than wild-type mice, as assessed using staining with periodic acid-

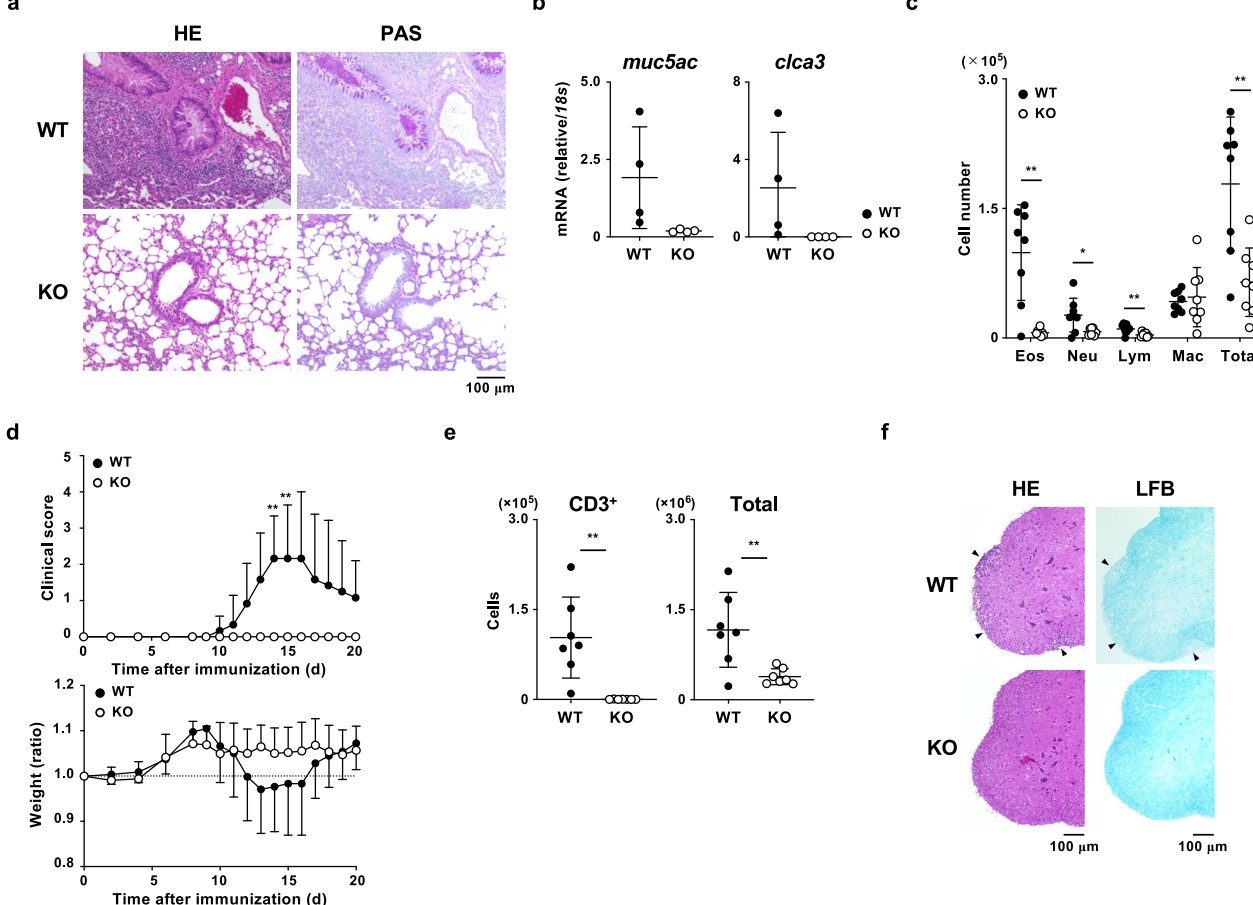

**Fig. 4 Attenuated helper T-cell-dependent inflammatory responses in Pgam1 KO mice. a–c** WT and T-cell-specific *Pgam1* KO mice were immunized intraperitoneally with 100 μg OVA in 2 mg of aluminum hydroxide gel. The immunized mice were intranasally challenged with OVA in saline (100 μg per mouse) on days 7, 8, and 9 to induce allergic airway inflammation. Twenty-four hours after the last OVA challenge, the lung samples (**a**) and BAL fluid samples (**c**) were prepared. **a** The microscopic appearance of the lungs fixed and stained with hematoxylin and eosin (H&E; left panel) or periodic acid-Schiff (PAS) reagent (right panel). Original magnification ×200 (Scale bars = 100 μm). **b** The results of the quantitative RT-PCR analysis of the *muc5ac* and *clca3* mRNA in the lungs. The results are presented relative to the expression of *18s* rRNA. Each point represents an individual mouse. **c** Quantification of eosinophils, neutrophils, lymphocytes, macrophages, and total cells in the BAL fluid (*n* = 8 per group). Each point represents an individual mouse. The results are presented relative to the expression of *18s* rRNA. Each point represents an individual mouse. **d–f** WT and *Pgam1* KO mice were immunized with MOG$_{35-55}$ peptides in CFA and pertussis toxin to induce experimental autoimmune encephalomyelitis. **d** The clinical scores (upper panel) and weight change (lower panel) were indicated (*n* = 6–7 per group). **e** Mice were killed on day 17, and the number of CD3$^+$ and total cells in the spinal cord were analyzed by flow cytometry (*n* = 7 per group). **f** The microscopic appearance of the spinal cords stained with H&E (left panels; arrows highlight inflammatory foci) or Luxol fast blue (LFB; right panels; arrows highlight demyelinated foci). Scale bars = 100 μm. The results are indicated with the standard deviation. *$P < 0.05$, **$P < 0.01$ (Student's *t* test).

Schiff reagent (Fig. 4a). Decreased mucus hyper-production was confirmed by the reduction in *muc5ac* and *mclca3* mRNA in the lungs of *Pgam1* KO mice (Fig. 4b). The infiltration of inflammatory cells, including eosinophils, neutrophils, and lymphocytes, in the bronchoalveolar lavage (BAL) fluid was significantly reduced in OVA-challenged *Pgam1* KO mice compared to OVA-challenged wild-type mice (Fig. 4c and Supplementary Fig. 9a). The effect of Pgam1 deficiency on the T$_H$1/T$_H$17-mediated autoimmune responses was determined using an experimental autoimmune encephalitis (EAE) model in vivo. As shown in Fig. 4d, *Pgam1* KO mice did not develop any signs of disease, while all wild-type mice developed such signs. A reduction in the body weight was observed in wild-type mice but not in *Pgam1* KO mice (Fig. 4d). In addition, the infiltration of T cells (CD3ε-positive cells) into the spinal cord was not detected, and the number of infiltrated cells was decreased in *Pgam1* KO mice (Fig. 4e). The reduction of infiltrating cells was also confirmed by

hematoxylin and eosin staining of spinal code (Fig. 4f and Supplementary Fig. 9b).

To assess demyelination and axonal damage within the central nervous system of diseased mice, histological analyses of spinal cord sections were carried out. The extent of demyelination and loss of axons were detected by luxol fast blue staining. Immunized Pgam1 KO mice showed fewer regions of demyelination than wild-type mice (Fig. 4f and Supplementary Fig. 9b). These results demonstrate that T$_H$-mediated inflammation is dependent on the glycolytic activation of CD4 T cells.

**Glycolysis supports sustained the activation of mTORC1 signaling.** To further investigate the regulatory role of glycolysis in T-cell activation and differentiation, we assessed the activation status of mTORC1 signaling. The phosphorylation status of S6 (Ser235/236) and 4E-BP1 (Thr37/46) proteins, which are downstream targets of mTORC1, in *Pgam1* KO naïve CD8 T cells at 8

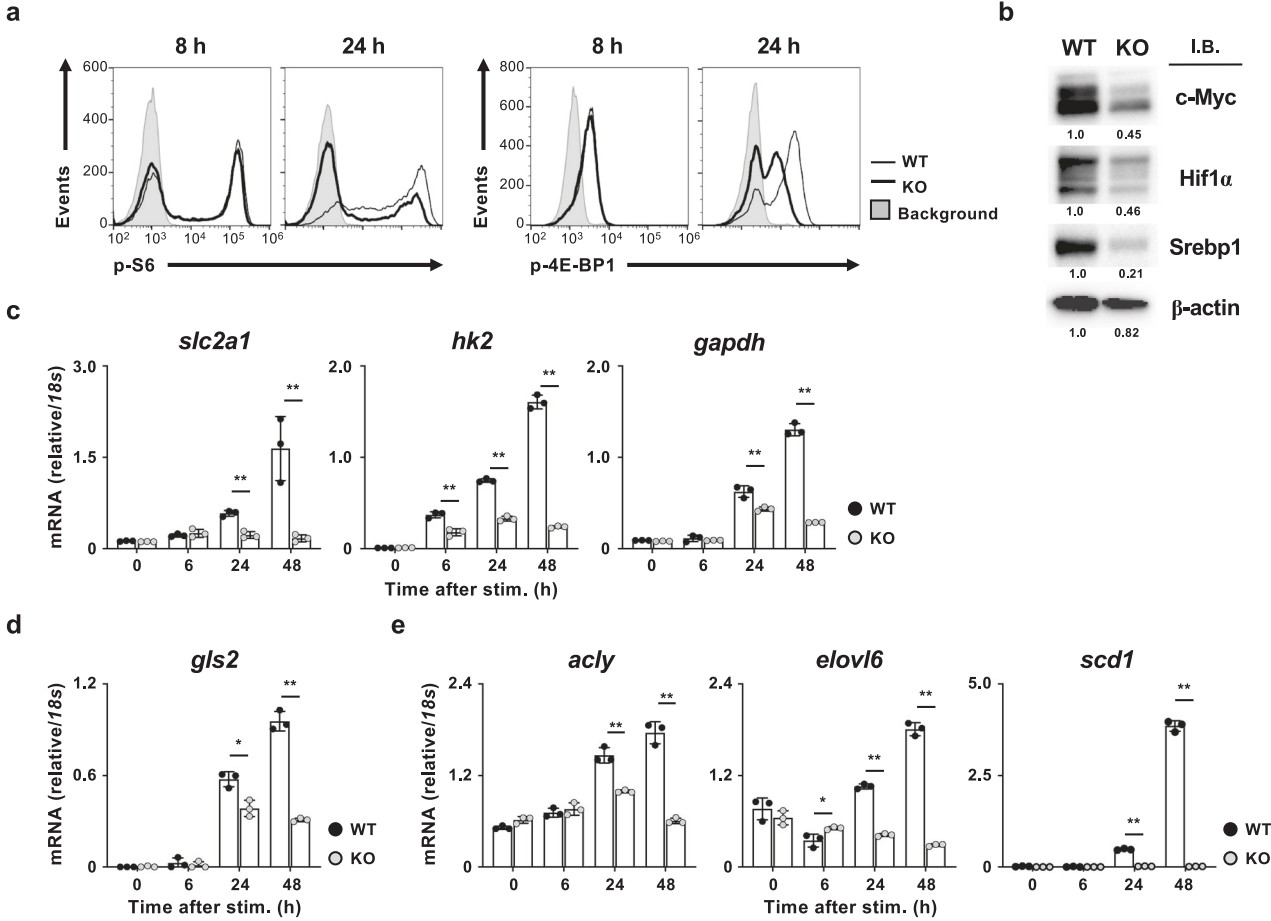

**Fig. 5 Glycolysis supports the sustained activation of mTORC1 signaling. a** Representative results of the intracellular FACS analysis of phospho-ribosomal S6 (S6; Ser235/236) and eukaryotic translation initiation factor 4E-binding protein 1 (4E-BP1; Thr37/46) in WT and *Pgam1* KO CD8 T cells stimulated with anti-TCR-β mAb plus anti-CD28 mAb for the indicated number of hours. The results of the FACS analyses are representative of at least three independent experiments with similar results. **b** The results of the immunoblot analysis of c-myc, Hif1α, SREBP1 and β-actin in WT and *Pgam1* KO CD8 T cells stimulated with anti-TCR-β mAb plus anti-CD28 mAb for 24 h. The protein amount of β-actin was used as a loading control. The numbers below the bands indicate the densitometry ratio with WT. The results of the immunoblot analyses are representative of at least three independent experiments with similar results and are presented as cropped images. The full-length blots are presented in Supplementary Figure 18. **c–e** WT and *Pgam1* KO CD8 T cells were stimulated with anti-TCR-β mAb plus anti-CD28 mAb for the indicated number of hours, and the mRNA expression of the enzymes for glycolysis (**c**), glutaminolysis (**d**), and lipid synthesis (**e**) were determined by qRT-PCR. The results are presented relative to the expression of *18s* rRNA with the standard deviation ($n = 3$, technical replicates). *$P < 0.05$, **$P < 0.01$ (Student's t test).

h after TCR stimulation was equivalent to that in wild-type CD8 T cells (Fig. 5a). However, the phosphorylation levels of both proteins were lower in *Pgam1* KO CD8 T cells at 24 h than in wild-type cells (Fig. 5a). Treatment with 2-DG showed similar effects on the phosphorylation of S6 (Ser235/236) protein in wild-type CD8 T cells (Supplementary Fig. 10). mTORC1 controls the protein expression of c-Myc, Hif1α and Srebp1 through post-transcriptional regulation[18]. As expected, the protein expression of c-Myc, Hif1α, and Srebp1 was decreased in *Pgam1* KO CD8 T cells 24 h after TCR stimulation (Fig. 5b). The levels of *c-Myc*, *Hif1α*, and *Srebp1* mRNA were marginally affected by Pgam1 deficiency (Supplementary Fig. 11a). mTORC1 provokes the expressions of enzymes and transporters that activate glycolysis, glutaminolysis, and fatty acid synthesis via the induction of c-Myc, Hif1α, and Srebp1. The induction of a facilitative glucose transporter *slc2a1*, *hk2*, and *gapdh* was lower in *Pgam1* KO activated CD8 T cells than in wild-type activated CD8 T cells (Fig. 5c). Furthermore, the decreased expression of *ldha* was detected in *Pgam1* KO activated CD8 T cells, while the levels of *ldhb* mRNA were increased at 24 and 48 h, but not 6 h, after TCR stimulation (Supplementary Fig. 11b). The reduction in the

*ldha/ldhb* ratio indicates the attenuation of anaerobic glycolysis in *Pgam1* KO activated CD8 T cells (Supplementary Fig. 11b). The mRNA expression of the regulatory targets of c-Myc (*gls2*) and Srebp1 (*acly*, *elovl6*, and *scd1*) was also attenuated in *Pgam1* KO activated CD8 T cells (Fig. 5d, e). These results suggest that TCR-mediated activation of glucose metabolism supports the sustained mTORC1 activation and induces the metabolic reprograming of T cells that is required for clonal expansion and differentiation.

**Glutamine is required for the sustained activation of mTORC1.** Next, we explored the molecular mechanism by which glycolysis induces the sustained activation of mTORC1 signaling. We previously reported that glutamine activates mTORC1 via the glutaminolysis-dependent supplementation of α-KG[27]. The levels of glutamine and glutamate were lower in *Pgam1* KO activated CD8 T cells in comparison to wild-type activated CD8 T cells at 24 h after TCR-stimulation, although those levels at 6 h were equivalent (Fig. 6a). In addition, the glutamine level in the culture medium was higher in *Pgam1* KO activated CD8 T cells in comparison to wild-type activated CD8 T cells at 24 h after TCR-stimulation, although those levels were equivalent at 6 h

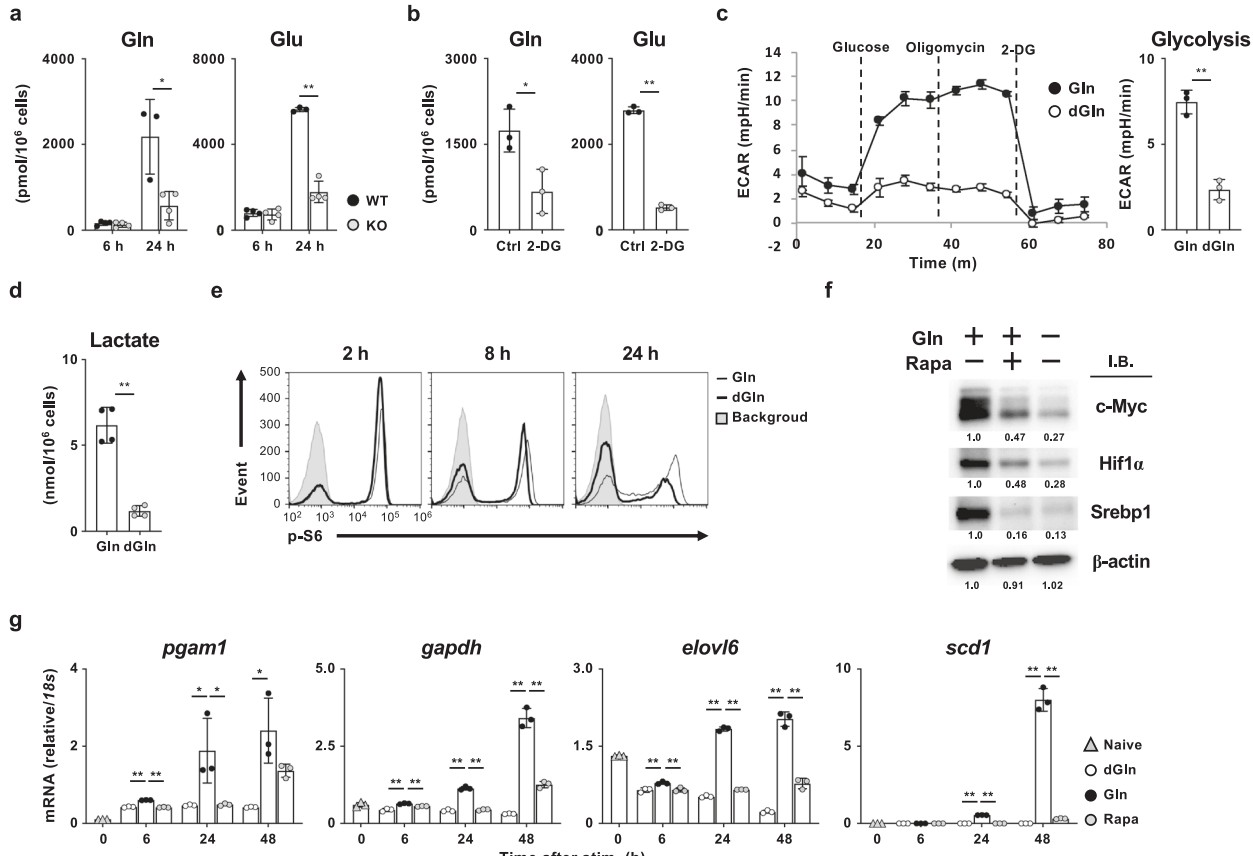

**Fig. 6 The glycolysis-dependent glutamine uptake is required for the sustained activation of mTORC1 signaling. a** The intracellular amounts of glutamine and glutamate in WT and *Pgam1* KO CD8 T cells stimulated with anti-TCR-β mAb plus anti-CD28 mAb for 6 or 24 h (*n* = 3–4, biological replicate). **b** The intracellular amounts of glutamine and glutamate in WT CD8 T cells stimulated with anti-TCR-β mAb plus anti-CD28 mAb in the presence or absence of 10 mM 2-DG for 24 h (*n* = 3, biological replicate). **c** WT naïve CD8 T cells were stimulated in the presence or absence of glutamine for 36 h, and then the ECAR was determined. The results are indicated with the standard deviation (right panel; *n* = 3, technical replicate). **d** The intracellular levels of lactate in activated CD8 T cells in the presence or absence of glutamine for 24 h are indicated with the standard deviation (*n* = 3, biological replicate). **e** Representative results of the intracellular FACS analysis of phospho-S6 (Ser235/236) in CD8 T cells stimulated in the presence or absence of glutamine for the indicated number of hours. **f** The results of the immunoblot analysis of c-Myc, Hif1α, Srebp1, and β-actin in CD8 T cells cultured in the presence or absence of glutamine and rapamycin for 24 h. The protein amount of β-actin was used as a loading control. The results are presented as cropped images. The numbers below the bands indicate the densitometry ratio with CD8 T cells cultured under Gln (+), Rapa (−) conditions. The results of the immunoblot analyses are representative of at least three independent experiments with similar results. The full-length blots are presented in Supplementary Fig. 18. **f** The levels of *Pgam1*, *Gapdh*, *Scd1*, and *Elovl6* mRNA in CD8 T cells stimulated in the presence or absence of glutamine for the indicated number of hours were determined by the quantitative RT-PCR analysis. The effect of rapamycin on glutamine-dependent alteration was also determined. The results are presented relative to the expression of *18s* rRNA with the standard deviations (*n* = 3, technical replicates). The results of the FACS analyses are representative of at least three independent experiments with similar results. *$p < 0.05$, **$p < 0.01$ (Student's *t* test).

(Supplementary Fig. 12). The intracellular concentrations of several amino acids (His, Ile, Leu, Lys, and Tyr) were moderately affected by Pgam1 deficiency (Supplementary Fig. 13). The reduction of intracellular glutamine and glutamate levels at 24 h after TCR stimulation was also observed by 2-DG treatment (Fig. 6b). These results indicate that glycolysis supports the increased glutamine uptake and subsequent glutaminolysis at 24 h after TCR stimulation in CD8 T cells.

Increases in the intracellular concentration of glutamine and glutamate in activated CD8 T cells were dependent on extracellular glutamine, since the levels of both amino acids were markedly decreased by the withdrawal of extracellular glutamine (Supplementary Fig. 14a). Glycolysis assessed by ECAR was decreased by the restriction of extracellular glutamine (Fig. 6c). The intracellular concentrations of glycolytic intermediates (F16P, DHAP, glycerophosphate, 3PG, and lactate), acetyl-CoA, and intermediates of TCA cycle (citrate, succinate, fumarate, and

malate) were decreased by the depletion of extracellular glutamine in the culture medium (glutamine-deprived conditions; Fig. 6d, Supplementary Fig. 14b, c). NADH/NAD$^+$ ratio in activated CD8 T cells was severely reduced by extracellular glutamine deprivation (Supplementary Fig. 14d). The intracellular levels of nucleotides were also decreased by extracellular glutamine deprivation (Supplementary Fig. 14e). These results indicate that both the glucose and glutamine metabolisms were impaired by glutamine deprivation in activated CD8 T cells in vitro.

Next, the effect of glutamine deprivation on mTORC1 activation was assessed by the phosphorylation status of S6 and 4E-BP1 proteins as an index. The phosphorylation levels of S6 protein (Fig. 6e) and 4E-BP1 (Supplementary Fig. 15a) in activated CD8 T cells cultured under glutamine-deprived conditions were comparable to those cultured under glutamine-sufficient conditions at 2 and 8 h after TCR stimulation. However,

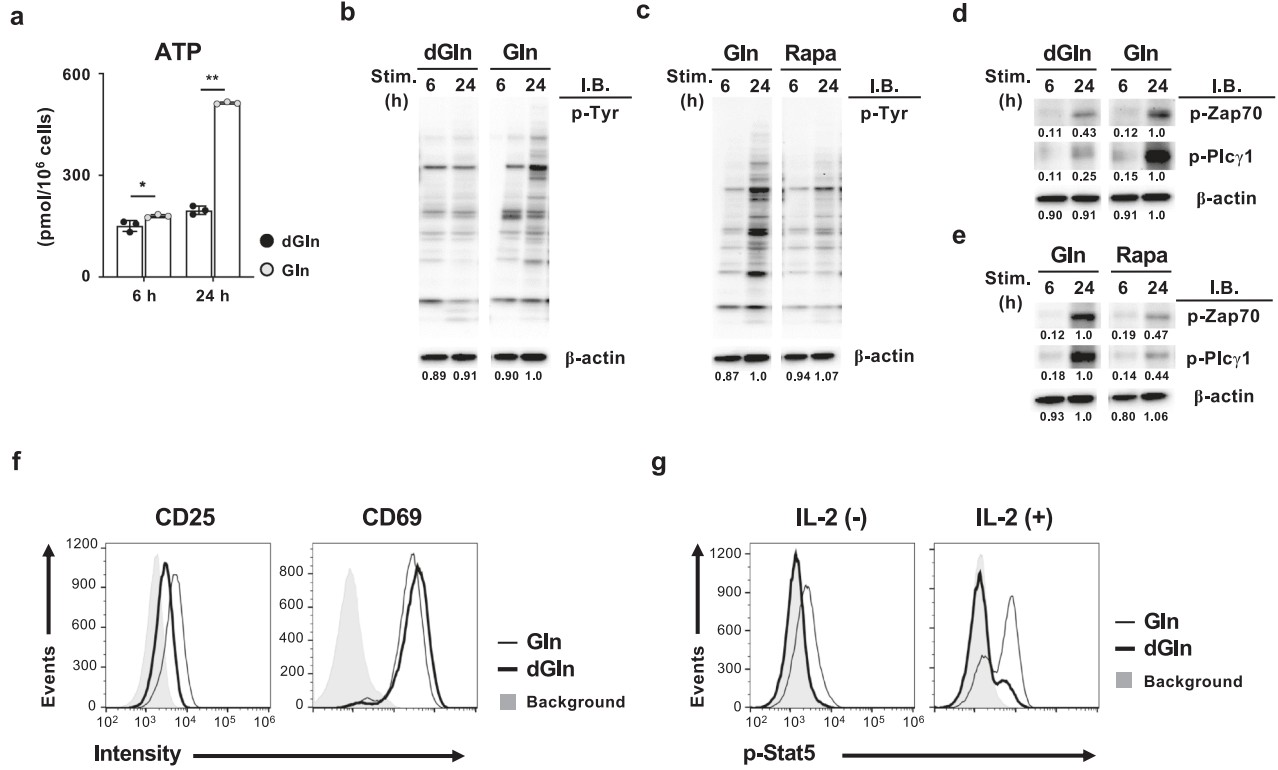

**Fig. 7 Glutamine controls TCR- and IL-2-mediated signaling in activated CD8 T cells. a** The intracellular amounts of ATP in CD8 T cells stimulated with anti-TCR-β mAb plus anti-CD28 mAb in the presence or absence of glutamine for the indicated number of hours. The results are presented with the standard deviations ($n = 3$, biological replicate). **b** The results of the immunoblot analysis of protein-tyrosine phosphorylation and β-actin in naïve CD8 T cells cultured with or without glutamine for the indicated number of hours. The protein amount of β-actin was used as a loading control. The numbers below the bands indicate the densitometry ratio with CD8 T cells cultured with glutamine. **c** The effect of rapamycin on glutamine-induced protein-tyrosine phosphorylation. Naïve CD8 T cells were stimulated under glutamine-sufficient conditions in the presence or absence of rapamycin for the indicated number of hours. The numbers below the bands indicate the densitometry ratio with CD8 T cells cultured without rapamycin. **d** The results of the immunoblot analysis of phospho-Zap70 (Tyr319), phosphor-Plcγ1 (Tyr783) and β-actin (control) in cells in (b). The numbers below the bands indicate the densitometry ratio with CD8 T cells cultured with glutamine. **e** The results of the immunoblot analysis of phospho-Zap70 (Tyr319), phosphor-Plcγ1 (Tyr783) and β-actin (control) in cells in (**c**). The numbers below the bands indicate the densitometry ratio with CD8 T cells cultured without rapamycin. **f** Representative results of the FACS analysis of CD25 and CD69 in CD8 T cells stimulated under the indicated conditions for 48 h. **g** Representative results of the intracellular FACS analysis of phosphor-Stat5 in CD8 T cells cultured with or without IL-2 stimulation. Naïve CD8 T cells were stimulated with anti-TCR-β mAb plus anti-CD28 mAb for 48 h, and then the cells were cultured with or without IL-2 for 24 h. The results of the FACS and immunoblot analyses are representative of at least three independent experiments with similar results. The results of the immunoblot analyses are presented as cropped images. The full-length blots are presented in Supplementary Fig. 18. *$p < 0.05$, **$p < 0.01$ (Student's $t$ test).

the phosphorylation levels of these proteins at 24 h after stimulation were lower in activated CD8 T cells cultured under glutamine-deprived conditions than in those cultured under glutamine-sufficient conditions (Fig. 6e and Supplementary Fig. 15a). Furthermore, the phosphorylation of S6 and 4E-BP1 proteins at 24 h after TCR stimulation was induced in a glutamine dose-dependent manner (Supplementary Fig. 15b). Consistent with the mTORC1 activity, the protein expression of c-Myc, Hif1α, and Srebp1 at 24 h after TCR stimulation was induced by glutamine and inhibited by rapamycin (Fig. 6f). In addition, the mRNA expression of *pgam1*, *gapdh*, *scd1*, and *elovl6* was induced in a glutamine-dependent manner in activated CD8 T cells and inhibited by rapamycin treatment (Fig. 6g). These results suggest that glutamine is required for the sustained activation of the mTORC1 signaling pathway in activated CD8 T cells.

**Glutamine controls TCR- and IL-2-mediated signals.** Finally, we examined whether or not glutamine regulates TCR- and IL-2-mediated signals in activated CD8 T cells, since these signals are impaired in *Pgam1* KO activated CD8 T cells. The intracellular

concentration of ATP was increased in a glutamine-dependent manner at 24 h after TCR stimulation. In contrast, the intracellular ATP concentration at 6 h after activation was less affected by glutamine (Fig. 7a). The protein-tyrosine phosphorylation in activated CD8 T cells was increased by extracellular glutamine at 24 h after TCR stimulation (Fig. 7b), and the glutamine-dependent upregulation of tyrosine-phosphorylation was inhibited by rapamycin (Fig. 7c). The tyrosine phosphorylation at 6 h after TCR stimulation was glutamine-independent (Fig. 7b) and rapamycin-insensitive (Fig. 7c). The tyrosine phosphorylation of the TCR signaling molecules Zap70 (Tyr319) and Plcγ1 (Tyr783) at 24 h after TCR stimulation was also enhanced by glutamine, and the increased tyrosine phosphorylation was inhibited by rapamycin (Fig. 7d, e). The phosphorylation of Zap70 and Plcγ1 at 6 h after TCR stimulation was induced in a glutamine-independent and rapamycin-insensitive manner (Fig. 7d, e).

We showed that glucose metabolism controls IL-2 signaling by regulating the CD25 expression (Fig. 2a, b). Therefore, we investigated the glutamine dependency of IL-2 signaling in activated CD8 T cells. The expression of CD25 was decreased under glutamine-deprived conditions, whereas the CD69

expression was not influenced (Fig. 7f). Furthermore, the IL-2-induced tyrosine phosphorylation of Stat5 was impaired in activated CD8 T cells cultured under glutamine-deprived conditions (Fig. 7g). Consistent with these results, the effector cytokine production in CD8 T cells cultured under glutamine-deprived conditions was markedly decreased compared with that in cells cultured under glutamine-sufficient conditions (Supplementary Fig. 16a, b). These results suggest that glutamine is required to sustain TCR- and IL-2-mediated signaling as well as glycolysis (Supplementary Fig. 17).

## Discussion

The mTORC1-mediated regulation of c-Myc, Hif1α, and Srebp1 protein expression has been reported[19,30]. In this study, we demonstrated that glutamine is required for c-Myc, Hif1α, and Srebp1 protein expression and sustained activation of mTORC1 signaling. Glutamine-induced c-Myc, Hif1α, and Srebp1 protein expression and the subsequent mRNA expression of glycolytic enzymes in activated CD8 T cells was inhibited by rapamycin, indicating that glutamine controls glycolysis through the regulation of the mTORC1 activity. The uptake of L-glutamine is the rate-limiting step in the essential amino acid- and growth factor-dependent activation of mTORC1. The counter transport of L-glutamine and L-leucine through the solute carrier family 7 member 5 (SLC7A5)/SLC3A2 complex controls mTORC1 activity[31]. Furthermore, we and others have reported that glutamine activates mTORC1 signaling via the supplementation of αKG[27,32]. We also found that the TCR-mediated glycolytic activation is required to sustain mTORC1 activity and to regulate the glutamine uptake in activated CD8 T cells. The inhibitions of glycolysis, glutamine uptake resulted in the impaired proliferation and effector differentiation of T cells. A previous report revealed that c-Myc sorting occurred asymmetrically during the first division[33]. c-Myc[low] cells do not show metabolic reprogramming or proliferative burst. The c-Myc-dependent regulation of the SLC1A5 expression and sustained mTORC1 activity was also reported[8,33]. Conversely, the transport of glutamine through SLC7A5, but not glutaminolysis, regulates the c-Myc expression in NK cells[34]. We demonstrated that the c-Myc protein expression was reduced by both the deletion of the Pgam1 and glutamine-depletion in activated CD8 T cells. The glutamine-dependent expression of the c-Myc protein was rapamycin-sensitive. Thus, these findings suggested that mTORC1, glycolysis, glutamine, and c-Myc affect each other and cooperate to induce proliferation, differentiation, and a T-cell-dependent immune response. Thus, the loss of Pgam1 influences the T-cell-dependent immune response through multiple pathways.

We showed in the present study that TCR signaling assessed by the tyrosine phosphorylation levels of Zap70 and Plcγ was not augmented in Pgam1 KO activated CD8 T cells. TCR-mediated protein-tyrosine phosphorylation did not show a time-dependent increase in Pgam1 KO naïve CD8 T cells. Glutamine was required for the augmentation of the TCR signaling and protein-tyrosine phosphorylation in activated CD8 T cells, and the effect of glutamine was antagonized by rapamycin. The upregulation of the intracellular ATP was also impaired by Pgam1 deficiency and glutamine restriction. These results suggest that both glycolysis and glutamine are required to activate signal transduction in activated T cells via the supplementation of metabolic products, such as ATP. Furthermore, ATP regulates the activity of ATP-dependent chromatin remodelers that control cellular differentiation[35]. ATP-dependent chromatin alteration is accomplished by chromatin-remodeling complexes that use the energy derived from ATP hydrolysis to alter nucleosome structure and

conformation. Changes in nucleosomes control transcription by regulating the accessibility of transcription factors[36]. Metabolic products, such as acetyl-CoA, αKG, S-adenosylmethionine, NAD+, succinate, and fumarate, are known to be used as substrates and cofactors of enzymes that regulate epigenetic modifications[37]. It is possible that mTORC1, glycolysis, and glutamine also control T cell differentiation by regulating cellular signals and the epigenetic status.

Pgam1 is reported to mediate PPP and serine synthesis through 3-PG and 2-PG in cancer cells[10], however, Pgam1 KO CD8 T cells did not show the reduction of PPP intermediates and serine. Although the causes of these inconsistency need further elucidation, the accumulation of upstream metabolites, glucose-6-phosphate and 3-PG, may diminish those effects. On the other hand, the intracellular nucleotide concentration strikingly decreased in Pgam1 KO CD8 T cells after 24 h from stimulation, despite the normal PPP activity. This is thought to be caused by the reduction of intracellular glutamine and ATP, which are essential for de novo nucleotide synthesis from ribose-5-phosphate. To support this observation, glutamine-deprivation also reduced the intracellular nucleotide concentration in CD8 T cells.

Pgam1 deficiency reduced OXPHOS in both CD4 and CD8 T cells at 24 h after TCR-stimulation. Although activated T cells dominantly rely on aerobic glycolysis to generate ATP, OXPHOS also increases activity during metabolic reprograming after TCR-stimulation in an mTOR-dependent manner[38,39]. Thus, it is hypothesized that the decreased mTORC1 activity, due to glycolytic failure, attenuated OXPHOS in Pgam1 KO T cells. Interestingly, Pagm1 KO T cells did not respond to proton ionophore FCCP at 24 h after stimulation. NADH and succinate are required to generate a proton gradient across the mitochondrial membrane via mitochondria respiratory complex I and II, respectively. The NADH and NAD+ levels were both decreased in Pgam1 KO activated CD8 T cells. The decreased purine nucleotide level possibly resulted in the reduction of the NADH and NAD+ levels. In addition, the intracellular levels of succinate and fumarate were also decreased in Pgam1 KO activated CD8 T cells. It is possible that the reduction of NADH and succinate, which are required to generate a proton gradient across the mitochondrial membrane, limits electron transport chain activity in Pgam1 KO CD8 T cells and causes poor responsiveness to FCCP.

The TCR-dependent expression of CD25 was inhibited by Pgam1 deficiency. The IL-2 responsiveness assessed by tyrosine phosphorylation of Stat5 was impaired in Pgam1 KO activated CD8 T cells. Furthermore, the expression of CD25 was low in activated CD8 T cells cultured under glutamine-restricted conditions compared to that in cells cultured under glutamine-sufficient conditions. The decreased expression of CD25 in Rptor, the scaffolding protein of mTORC1 complex, -deficient T[H]2 cells was previously reported[19]. Similar to that observed in Pgam1 KO activated T cells, the expression of CD44 and CD69 was unchanged in Rptor KO T cells. In addition to the Rptor and Pgam1 KO activated CD8 T cells, the mTORC1 activity was also abrogated in activated CD8 T cells under glutamine-restricted conditions. These results suggest that the responsiveness to IL-2 and the proliferative response of activated CD8 T cells are dependent on mTORC1, glycolysis, and glutamine, all of which are indispensable.

The differentiation of T[H]1, T[H]2, and T[H]17 was impaired in Pgam1 KO naïve CD4 T cells, whereas the Foxp3-positive Treg cell generation was only marginally affected. The T[H]17/Treg ratio was reduced in Pgam1 KO CD4 T cell under T[H]17 conditions compared to wild-type CD4 T cells. These results were consistent with a previous observation in which effector T cell development

was found to depend on anaerobic glycolysis, while Treg cells preferentially used β-oxidation[40–43]. However, the reduction of IFN-γ-producing $T_H1$ cells in *Pgam1*-deficient CD4 T cells was milder than that in IL-4-producing $T_H2$ cells and IL-17-producing $T_H17$ cells in our experiments. In addition, the reduced generation of IFN-γ-producing $T_H1$ cells in *Pgam1*-deficient CD4 T cells was restored by the supplementation of pyruvate in vitro, whereas the development of $T_H2$ and $T_H17$ cells was not. We demonstrated that OXHOS was reduced in *Pgam1*-deficient activated CD4 T cells; thus, it is possible that OXPHOS also plays an important role in $T_H1$ differentiation. These results suggest that glycolysis induces the differentiation of $T_H$ subsets through various different pathways.

Several studies have shown that the generation of Treg can be enhanced by rapamycin in vitro and in vivo[44–46]. Furthermore, the genetic deletion of components of the mTORC signaling pathway in T cells results in the enhanced generation of Tregs and decreased effector T cell development[47,48]. However, Treg-specific *Rptor* KO mice developed systemic autoimmunity, suggesting that the mTORC1 activity is required for the Treg function[49]. The number of Tregs was decreased in the spleen of *Pgam1* KO mice, and we did not examine the function of *Pgam*-deficient Treg cells. Recently, the mTORC2-dependent $T_H1$, $T_H2$ and Treg differentiation and functions have been reported[50,51]. In the present study, we did not assess the effect of Pgam1 deficiency or glutamine-deprivation on the mTORC and Akt activity in CD4 T cells. It is likely that glycolysis and glutamine also regulate the mTORC2 activity via the supplementation of ATP. Thus, the interaction between mTOR, glycolysis, and glutamine in CD4 T cells should be further clarified in the future.

## Methods

**Mice**. CD4-Cre transgenic (TG) mice, OT-1 Tg mice, Thy1.1$^+$ mice, and Thy1.2$^+$ mice were purchased from The Jackson Laboratory (Bar Harbor, ME, USA). *Pgam1*$^{flox/flox}$ mice were kindly provided by Drs. Hiroshi Kondoh and Takumi Mikawa (Kyoto University). In brief, the exon 2–4 of pgam1 was pinched by two loxP sequences, and FLP-flanked neo-cassette was inserted in the 5′ upstream region of exon 2 to generate the pgam1 targeting vector. Targeting vector was injected into C57BL/6 mouse embryonic stem cells. The founder mice were crossed with *FLP* transgenic mice to remove the neo-cassette gene to generate Pgam1-floxed mice. Pgam1-floxed mice were crossed with CD4-Cre TG mice to generate T-cell-specific Pgam1-deficient mice. Gene-manipulated mice with a C57BL/6 background were used in all experiments. These mice were purchased from Clea Japan, Inc. (Tokyo, Japan). Both male and female mice were used in the experiments. All mice were used at 8–12 weeks of age.

All of the animal experiments received approval from the Ehime University Administrative Panel for Animal Care. All animal care was conducted in accordance with the guidelines of Ehime University. All experiments using *Lm*-OVA were performed in accordance with the protocols approved by the Ehime University Institution Biosafety Committee.

**Reagents and antibodies**. Rapamycin was purchased from Wako Chemicals (cat# R0161; Osaka, Japan). Antibodies used for intracellular and cell-surface staining were as follows: anti-CD8-violetFluor 450 mAb (0.066 μg/10$^6$ cells, 2.43; TONBO Biosciences, San Diego, CA, USA), anti-CD4-fluorescein isothiocyanate (FITC), mAb (0.04 μg/10$^6$ cells, cat# 553047; BD Biosciences, San Jose, CA, USA), anti-TNF-α-phycoerythrin (PE), mAb (0.08 μg/10$^6$ cells, cat# 554419; BD Biosciences), anti-IFN-γ-FITC mAb (0.025 μg/10$^6$ cells, cat# 554411; BD Biosciences), IL-2-allophycocyanin (APC), mAb (0.01 μg/10$^6$ cells, cat# JES6-5H4; TONBO Biosciences), anti-Phospho-S6 (Ser235/236)-Alexa Fluor 647 mAb (0.125 μg/10$^6$ cells, cat# 4851; Cell Signaling Technology, Danvers, MA, USA), anti-4E-BP1 (Thr37/46)-Alexa Fluor 488 mAb (0.05 μg/10$^6$ cells, 236B4; cat# 2846 S; Cell Signaling Technology), anti-phospho-STAT5 (Tyr694)-Alexa Fluor 647 mAb (20 μl/test/10$^6$ cells, cat# 612599; BD Biosciences), anti-CD44-PE mAb (0.02 μg/10$^6$ cells, IM7; TONBO Biosciences), anti-CD62L-APC mAb (0.02 μg/10$^6$ cells, cat# 20-0621; TONBO Biosciences), anti-CD25-PE mAb (0.066 μg/10$^6$ cells, 3C7; BD Biosciences), anti-CD40L-PE mAb (0.02 μg/10$^6$ cells, cat# 106505; BioLegend), anti-Thy1.1-Alexa Fluor 647 mAb (0.005 μg/10$^6$ cells, OX-7; BioLegend), anti-Thy1.2-APC-Cy7 mAb (0.02 μg/10$^6$ cells, 30-H12; BioLegend), anti-IL-4-PE mAb (0.066 μg/10$^6$ cells, cat# 554435; BD Bioscience), anti-IL17A-PE mAb (0.066 μg/10$^6$ cells, cat# 506904; BioLegend), and anti-Foxp3- Alexa Fluor 647 mAb (0.25 μg/10$^6$ cells, cat# 320014; BioLegend).

The antibodies for immunoblotting were as follows: anti-phospho-tyrosin mAb (1:1000, cat# 8954 S; Cell Signaling Technology), anti-phospho-zap70 (Y319/352) mAb (1:1000, cat# 2717 T; Cell Signaling Technology), anti-phospho-PLCγ1 (Y783) mAb (1:1000, cat# 14008 T; Cell Signaling Technology), anti-β-actin Ab (1:1000, cat# 4970SS; Cell Signaling Technology), anti-c-Myc mAb (1:1000, cat# 5605 S; Cell Signaling Technology), anti-Hif1α Ab (1:2000, A300-286A; Bethyl Laboratories, Montgomery, AL, USA), and anti-Srebp1 mAb (1:200, cat# 13551; Santa Cruz Biotechnology, Dallas, TX, USA). All antibodies were diluted and used according to the manufacturer's instructions.

**T-cell stimulation and differentiation in vitro**. Naive CD8 T (CD44$^{low}$CD62L-$^{high}$) cells and naive CD4 T (CD44$^{low}$CD62L$^{high}$CD25$^{negative}$) cells were prepared using a Naive CD8$^+$ T-cell Isolation kit (cat# 130-096-543; Miltenyi Biotec, San Diego, CA, USA) and a Naive CD4$^+$ T-cell Isolation kit (cat# 130-093-227; Miltenyi Biotec), respectively. Naive T cells (7.5 × 10$^5$) were stimulated with immobilized anti-TCR-β mAb (3 μg/ml, H57-597; BioLegend) and anti-CD28 mAb (1 μg/ml, 37.5; BioLegend) for 2 days. The cells were then transferred to a new plate and further cultured.

The cytokine conditions were as follows: IL-2 conditions, IL-2 (10 ng/ml); Th1 conditions, IL-2 (10 ng/ml), IL-12 (1 ng/ml), and anti-IL-4 mAb (5 mg/ml, 11B11; BioLegend); Th2 conditions, IL-2 (10 ng/ml), IL-4 (1 ng/ml), and anti-IFN-γ mAb (5 mg/ml, R4-6A2; BioLegend); Th17 conditions, IL-6 (10 ng/ml), IL-1β (10 ng/ml), TGF-β1 (1 ng/ml), anti-IL-4 mAb (5 mg/ml), and anti-IFN-γ mAb (5 mg/ml). The cells were cultured in RPMI 1640 with L-glutamine (cat# 189-02025; Wako Chemicals) supplemented with 10% fetal bovine serum, 2 mM L-glutamine (16948-04; Nacalai Tesque, Kyoto, Japan), 1% MEM nonessential amino acids (cat# 06344-56; Nacalai Tesque), 10 mM HEPES (cat# 15630-080; Thermo Fisher Scientific, Waltham, MA, USA), 55 μM 2-Mercaptoethanol (cat# 21985-023; Thermo Fisher Scientific), and 1% penicillin-streptomycin (cat# 26253-84; Nacalai Tesque).

For glutamine-deprived conditions, the cells were cultured in RPMI 1640 without L-glutamine (cat# 183-02165; Wako Chemicals) supplemented with 10% fetal bovine serum, 1% MEM nonessential amino acids, 10 mM HEPES, 55 μM 2-mercaptoethanol, and 1% penicillin-streptomycin. The medium was basically deprived of pyruvate, an end product of the glycolytic pathway.

**Intracellular staining of cytokines**. The cells were differentiated in vitro and stimulated with an immobilized anti-TCR-β mAb (3 μg/ml, H57-597; BioLegend) for 6 h with monensin (2 μM, cat# M5273; Sigma-Aldrich, St. Louis, MO, USA) for the intracellular staining of cytokines. Intracellular staining was then performed as described previously[27,52]. For the intracellular staining of phosphorylated S6 ribosomal proteins, the CD8 T cells were fixed and permeabilized with BD Phosflow Lyse/Fix Buffer (cat# 558049; BD Biosciences) and BD Phosflow Perm III (cat# 558050; BD Biosciences) in accordance with the manufacturer's instructions. Flow cytometry (FACS) was performed using a FACS Caliber instrument (BD Biosciences) and Gallios instrument (Beckman Coulter, CA, USA), and the results were analyzed using the FlowJo software program (Tree Star, Ashland, OR, USA).

**ELISA**. The cells were stimulated with an immobilized anti-TCR-β mAb (3 μl/ml) for 16 h. The concentrations of TNF-α, IL-13, IL-17A, and IL-17F were determined using the DuoSet ELISA kits (R&D Systems, Minneapolis, MN, USA) in accordance with the manufacturer's instructions. The concentrations of IL-2, IFN-γ, IL-4, and IL-5 in the supernatants were determined using ELISAs, as described previously[52].

**Quantitative reverse transcriptase polymerase chain reaction**. Total RNA was isolated using TRIzol regent, and complementary DNA (cDNA) was synthesized using the superscript VILO cDNA synthesis kit (cat# 11754; Thermo Fisher Scientific). Quantitative reverse transcriptase polymerase chain reaction (qRT-PCR) was performed using the Step One Plus Real-Time PCR System (Thermo Fisher Scientific).

**Immunoblot analyses**. Nuclear and cytoplasmic extracts were prepared using NE-PER Nuclear and Cytoplasmic Extraction Reagents (cat# 78833; Thermo Fisher Scientific). The lysates were separated on an SDS polyacrylamide gel and then subjected to immunoblotting with specific antibodies. The western blot bands were quantified using the Image J software program (http://rsb.info.nih.gov/ij).

**Metabolic profiling**. Metabolome measurements and data processing were performed through a facility service at Human Metabolome Technology, Inc. (Yamagata, Japan) as we described previously[27].

In brief, WT and *Pgam1* KO naïve CD8 T cells were stimulated with plate-bound anti-TCR-β mAb plus anti-CD28 mAb in the presence or absence of glutamine for 24 h. The cells (5 × 10$^6$ cells) were washed with 5% (w/w) mannitol and then lysed with 800 μl of methanol and 500 μl of Milli-Q water containing internal standards (H3304-1002; Human Metabolome Technology, Inc.) and left to rest for another 30 s. The extract was obtained and centrifuged at 2300 × *g* at 4 °C for 5 min, and then 800 μl of the upper aqueous layer was centrifugally filtered through a Millipore 5-kDa cut-off filter at 9100 × *g* at 4 °C for 120 min to remove proteins. The filtrate was centrifugally

concentrated and re-suspended in 50 µl of Milli-Q water for capillary electrophoresis-mass spectrometry (CE-MS). Cationic compounds were measured in the positive mode of capillary electrophoresis-time of flight-mass spectrometry (CE-TOFMS), and anionic compounds were measured in the positive and negative modes of capillary electrophoresis-tandem mass spectrometry (CE-MS/MS), in accordance with the methods developed by Soga et al[53,54].

The ECAR and OCR were measured using an Extracellular Flux Analyzer XFp (Agilent Technologies, Santa Clara, CA, USA). The culture medium was changed to Seahorse XF RPMI Base Medium (Cat# 103336-100; Agilent Technologies) before the analysis. Activated CD8 T cells ($1.5 \times 10^5$ at 8 h and $1 \times 10^5$ at 24 h after activation) or CD4 T cells ($1.5 \times 10^5$) were adhered to Cell Tak coated seahorse eight-well plates and pre-incubated at 37 °C for 45 min in the absence of $CO_2$. The ECAR was then measured at the baseline and in response to 10 mM glucose, 1 µM oligomycin, and 50 mM 2-DG (XFp glycolysis stress test kit; cat# 103017-100; Agilent Technologies). The OCR was measured under basal conditions and in response to 1 µM oligomycin, 2 µM FCCP and 0.5 µM rotenone/antimycin A (XFp mito stress test kit; cat# 103010-100; Agilent Technologies).

**Adoptive transfer of CD8 T cells and Listeria infection**. Naïve CD8 T cells were prepared from the spleens of wild-type OT-1 Tg (Thy1.1$^+$) or Pgam1 KO OT-1 Tg mice (Thy1.2$^+$). The purified cells were mixed at a 1:1 ratio (wild-type/Pgam1 KO) and intravenously transferred into naive C57BL/6 (Thy1.1$^+$Thy1.2$^+$) mice ($1 \times 10^4$ cells per mouse). The mice were then infected with the Listeria monocytogenes-expressing OVA (Lm-OVA) strain at $5 \times 10^3$ colony-forming units (CFU) intravenously 18–24 h later. The donor cells in the spleen were gated, and the expansion was assessed on days 5 and 7 after Lm-OVA infection.

**EAE induction in mice**. Wild-type and T cell-specific Pgam1 KO mice were immunized subcutaneously with 200 µg MOG35-55 peptide in CFA. The mice were infected with pertussis toxin (100 ng) intraperitoneally on days 0 and 2, and monitored for the EAE clinical score and weight changes. Possible EAE scores were 0 = normal, 1 = flaccid tail, 2 = hind limb weakness, 3 = partial hind limb paralysis, 4 = complete hind limb paralysis, and 5 = moribund state. For FACS and histological analyses, mice were killed on day 17.

**OVA-induced allergic inflammation in mice**. Wild-type and T-cell-specific Pgam1 KO mice were immunized intraperitoneally with 100 µg OVA in 2 mg of aluminum hydroxide gel on day 0. Next, the mice were intranasally challenged with OVA in saline (100 µg per mouse) on days 7, 8, and 9. Twenty-four hours after the last OVA challenge, BAL-fluid cells and lung samples were prepared for a histological examination.

**Primers and probes for qRT-PCR**. The primer and TaqMan probes used for the detection of Gapdh were purchased from Thermo Fisher Scientific. The specific primers and Roche Universal Probes were as follows: acly: 5′-AAGAAGGAGG GGAAGCTGAT-3′ (forward), 5′-TCGCATGTCTGGGTTGTTTA-3′ (reverse), probe#105 c-myc: 5′-CCTAGTGCTGCATGAGGAGA-3′ (forward), 5′-TCCACA GACACCACATCAATTC-3′ (reverse), probe#77, elovl6: 5′-GCAAAGCACCCGA ACTAGG-3′ (forward), 5′-GAGACACAGTGATGTGGTGGT-3′ (reverse), probe #4, gls2: 5′-GTATGACTTCTCGGGCCAGT-3′ (forward), 5′-TCCTGACCCAGC TGACTTGG-3′ (reverse), probe #97, hif1a: 5′-GGCTATCCACATCAAAGCAA-3′ (forward), 5′-GCACTAGACAAAGTTCACCTGAGA-3′ (reverse), probe #95, hk2: 5′-CAACTCCGGATGGGACAG-3′ (forward), 5′-CACACGGAAGTTGGTTC CTC-3′ (reverse), probe #21, ldha: 5′-GGCACTGACGCAGACAAG-3′ (forward), 5′-TGATCACCTCGTAGGCACTG-3′ (reverse), probe #12, ldhb: 5′-GCCCTG AGTCCAGGAAGTAAG-3′ (forward), 5′-CGATGGCTCCTGTGAGTTTT-3′ (reverse), probe #63, pgam1: 5′-AGTTCCTGGGAGACGACGAG-3′ (forward), 5′-CTGCCTCTTCACCTTCACTTC-3′ (reverse), probe #5, scd1: 5′-TTCCCTCCTG CAAGCTCTAC-3′ (forward), 5′-CAGAGCGCTGGTCATGTAGT-3′ (reverse), probe #34, pgam2: 5′-AACCTCATGGGCTTGG-3′ (forward), 5′-GGCATTGT-GAAACATC -3′ (reverse), probe #93, scd1: 5′-TTCCCTCCTGCAAGCTCTAC-3′ (forward), 5′-CAGAGCGCTGGTCATGTAGT-3′ (reverse), probe #34, srebf1: 5′-GGYYYGAACGACATCGAAGA-3′ (forward), 5′-CGGGAAGTCACTGTCTT GGT-3′ (reverse), probe #78, slc2a1: 5′-GACCCTGCACCTCATTGGT-3′ (forward), 5′-GATGCTCAGATAGGACATCCAAG-3′ (reverse), probe #99, and 18s: 5′-GCAATTATTCCCCATGAACG-3′ (forward), and 5′-GGGACTTAATCAA CGCAAGC-3′ (reverse), probe #48.

**Statistics and reproducibility**. Student's t test was used for the statistical analyses. Experiments were independently repeated at least three times in general as mentioned in figure legends.

**Reporting summary**. Further information on experimental design is available in the Nature Research Reporting Summary linked to this paper.

## Data availability
The datasets generated during and/or analyzed during the current study are available from the corresponding author on reasonable request. Source data can be found in Supplementary Data 1.

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

## Acknowledgements

We thank A. Tamai and D. Shimizu for their excellent technical assistance. This work was supported by JSPS KAKENHI Grants (18K08409, 18H05036, 17H04086, 18K05364, 20H035040, and 20H049480), the Kanae Foundation for the Promotion of Medical Science, the Naito Foundation Natural Science Scholarship, the Uehara Memorial Foundation, and the TAKEDA Science Foundation.

## Author contributions

K.T. and M.K. performed the experiments, analyzed the data and wrote the manuscript, N.T., A.K., T.Yorozuya, T.Yamada, T.S., A.S. and M.Y. performed and supported the experiments, H.K. and T.M. established and provided the *Pgam1*^flox/flox^ mice, and M.Y. conceptualized the research, directed the study, and edited the manuscript.

## Competing interests

The authors declare no competing interests.
