## [Peer Review File · Communications Biology]

Reviewers' comments:

Reviewer #1 (Remarks to the Author):

Toriyama et al identified the critical role of glutamine-dependent glycolytic activation in regulating T-cell immune responses, which suggest that mTORC1, glycolysis and glutamine form an amplification loop to induce proliferation, differentiation and T cell-dependent immune responses. My comments are as below:

Major Points:

1. The authors generate T cell-specific Pgam1 KO mice in the first time, it is not clear whether Pgam1 KO in T cells impairs its initial function, it would be to do some experiments to test the cellular level of 3-PG and 2-PG in T cells.
2. In Fig.5, the data indicate that glycolysis supports sustained the activation of mTORC1 signaling, the level of the mTORC1 downstream targets, p-S6 and p-4E-BP1, at early 2 h after TCR stimulation was comparable, However, the phosphorylation levels of both proteins were lower in Pgam1 KO CD8 T cells at late 36 h than in wild-type cells. Meanwhile, Fig.1b shows that Pgam1 KO impairs CD8 T cells glycolysis in 36h. It is important to detect the glycolysis efficiency in Pgam1 KO cells in early time, such as 8h or 12h, to prove that glycolysis inhibition is prior to mTORC1 dysfunction.
3. The topic is about T cell immune response, and the deficiency of Pgam1 influenced the metabolism and proliferation of CD8 T cell. Since the CD8+ T cell is crucial for antitumor immunity, what is the role of Pgam1 in T cell-mediated tumor immunity? Or if there are some experiments about the influence of Pgam1 on CD8 T cell effector function?
4. The authors described the influence of Pgam1 on CD4 and CD8 T cell has some differences, but what is the mechanism of this?
5. According to paper's description, the level of glutamine is lower in Pgam1 KO activated CD8 T cell than in wild-type activated CD8 T cell (Fig. 6a), there is no significant difference between WT and KO. And the experiments after are all about glutamine. So if there are more evidence showed the relation between Pgam1 and glutamine?

Minor comments:

1. There is a minor mistake in figure legend of Supplementary Fig.1. There are two SFig.1d without Fig.1e which is not consistent with paper.
2. In Fig.1d, there is significant difference of ATP between WT and KO in 6h. Why the paper described that the ATP in 6h is comparable?
3. In the result of Impaired TH1, TH2 and TH17 cell differentiation in Pgam1 KO naïve CD4 T cells, according to the paper's description, the expression of CD40L (Supplementary Fig. S4a) is decreased, which is contrast to the results showed in Supplementary Fig. S4a, which showed marked increase.
4. For the phosphorylation of S6 and 4E-BP1, whether the stimulation for 36h is too long to test the phosphorylation status?

Reviewer #2 (Remarks to the Author):

Toriyama et al., present a mouse model of T-cell specific glycolytic impairment by T cell specific Pgam1 deletion. The authors show compelling in vitro and in vivo data from multiple models of functional T cell defects in the CD4 and CD8 subsets. By tracking antigen specific responses in vivo, the authors demonstrate that the Pgam1-deficient T cells fail to accumulate in infected animals, and show functional perturbations. The authors provide evidence to show that both the CD8 and CD4-Th17 responses are strongly inhibited, leading to 1) failure to clear infection efficiently, and 2) protection

against induction of IBD, respectively. To provide mechanistic insight, the authors performed basic metabolic and signaling characterization of the Pgam1-deficient and WT T cells in vitro and conclude that mTORC1 signaling, glycolysis and glutamine form an amplification loop to induce proliferation, differentiation and T cell-dependent immune responses.

Where the cellular and in vivo data of the mouse model as shown in figure 1-4 are compelling and novel, the link from this part of the manuscript to the following 3 figures is somewhat abrupt, leading into mechanisms that were not clearly linked back to the role of Pgam in T cells. Further elucidation of the role of Pgam and the metabolites handled by this enzyme in the T cell subsets identified in the in vivo models would greatly strengthen the manuscript (even from in vitro differentiated T cells). In addition, this reviewer believes that the manuscript as presented falls short to substantiate the mechanistic claims regarding the cross talk or amplification loop between mTORC1 signaling, glucose and glutamine metabolism and T cell function, and misses some references to relevant publications related to this cross talk.

Concerns:

1. The authors show compelling evidence that ECAR and OCR are affected by Pgam1 deletion, but these assays were performed at 36 hours after activation, right around the time that T cells start proliferating and asymmetrically inherit cMyc (Verbist et al., *Nature*. 2016 Apr 21; 532(7599): 389–393. not cited by the authors). Low cMyc cells do not develop into full effectors, display low CD25 expression (could explain lower STAT3 signaling), mTOR signaling, fail to sustain numbers in vivo, and do not undergo metabolic remodeling. In a different study it was shown that cMyc protein expression is sensitive to Slc1a5 inhibition (Glutamine transporter) in NK cells (Loftus et al., *Nat Commun*. 2018 Jun 14;9(1):234). This suggests that there is intricate two-way cross talk between the expression of cMyc, mTOR and glutamine availability in immune cells, and should be discussed, especially in light of the conclusions drawn.
2. The authors show that proximal signaling is intact during the first 2-6 hours of cells (figure 5), but the obvious changes happen at the moment that the cell cycle would be engaged at 36 hours. In light of this, how do previous findings as described under point 2 link to the current study? Is the perturbation of the cell cycle here the cause or the effect?
3. The use of Stat3 as a loading control for western blots seems to be an odd choice in differentially activated cells. Additionally, the loading based on Stat3 is obviously not equal in the blots shown in figures 6e, 7b, 7c, 7d and 7e, making the differences as shown hard to interpret. A more conventional loading control should be included that reflects either cell numbers lysed or total protein on the membrane.
4. The metabolic data as presented in figures 3-6 are lacking in addressing the complexity of the pathways impacted by Pgam1 deletion, and are focused on glutamine depletion, and treatment with the mTORC1 inhibitor rapamycin. The title of figure 6 states 'glycolysis-dependent glutamine import' but the data in figure 6a indicate no significant differences in glutamine, making the link to the figure title and glutamine depletion somewhat odd. The lower pools of glutamate in the KO cells could be the result of altered activity of GLS (glutamine to glutamate), altered assimilation of glutamate in the TCA cycle (via 2-OG), or the increased exchange of glutamate for the import of cystine (by xCT11) to sustain glutathione synthesis.
5. The authors highlight the function of Pgam1 in the introduction on page 4 as a mediator of glycolysis, the PPP and Serine synthesis. However, the 3-PG accumulation can inhibit the PPP in generating nucleotides (important for proliferation), serine was previously shown to be important for T cell proliferation and the generation of one-carbon units for nucleotide synthesis (Ma et al., *Cell Metab*.

2017 Feb 7;25(2):345-357),

As glutamine is a biosynthetic precursor and important nitrogen source important for nucleotide synthesis, depletion might overlap with the Pgam1 KO phenotype, but might not be the mechanistic equivalent. An informative assay to address this further could be the supplementation of Pgam1 KO T cells with cell permeable metabolic intermediates (such as dimethyl-modified TCA intermediates), or alternative 3-carbon substrates that bypass the block at 3-PG, such as alanine, pyruvate or lactate.

RESPONSES TO THE REVIEWERS

Reviewer #1

(Remarks to the Author):

Toriyama et al identified the critical role of glutamine-dependent glycolytic activation in regulating T-cell immune responses, which suggest that mTORC1, glycolysis and glutamine form an amplification loop to induce proliferation, differentiation and T cell-dependent immune responses. My comments are as below:

Appreciation to Reviewer #1

We especially appreciate this reviewer's critical reading and suggestions. Based on this reviewer's suggestions, we have performed several new experiments.

Major Points:

1. The authors generate T cell-specific Pgam1 KO mice in the first time, it is not clear whether Pgam1 KO in T cells impairs its initial function, it would be to do some experiments to test the cellular level of 3-PG and 2-PG in T cells.

Response: According the reviewer's suggestion, we performed metabolic profiling of *Pgam1* KO CD8 T cells at 6 h and 24 h after TCR-mediated stimulation. We detected an increased level of glycolytic intermediates that are located before the Pgam-dependent catabolizing step (G6P, F6P, F1-6P DHAP and 3PG) in *Pgam1* KO CD8 T cells at 24 h after TCR-mediated activation. In contrast, the intracellular level of lactate, an end-product of anaerobic glycolysis, was decreased in *Pgam1* KO cells. The results are shown in **Fig. 1b**. The results of metabolic profiling in *Pgam1* KO CD8 T are also indicated in **Fig. 6a**, **Supplementary Fig.S4c-4e** and **Supplementary Fig.S12**.

2. In Fig.5, the data indicate that glycolysis supports sustained the activation of mTORC1 signaling, the level of the mTORC1 downstream targets, p-S6 and p-4E-BP1, at early 2 h after TCR stimulation was comparable, However, the phosphorylation levels of both proteins were lower in Pgam1 KO CD8 T cells at late 36 h than in wild-type cells. Meanwhile, Fig.1b shows that Pgam1 KO impairs CD8 T cells glycolysis in 36h. It is important to detect the glycolysis efficiency in Pgam1 KO cells in

early time, such as 8h or 12h, to prove that glycolysis inhibition is prior to mTORC1 dysfunction.

Response: According to the reviewer's suggestion, we assessed the glycolysis efficiency in *Pgam1* KO CD8 T cells at 8 h (**Supplementary Fig.S4c**) and 24 h (**Fig. 1a**) after TCR stimulation. The glycolytic activity in *Pgam1* KO CD8 T cells at 24 h after TCR stimulation was lower than that in wild-type cells, whereas the level in *Pgam1* KO CD8 T cells at 8 h was comparable to that in wild-type cells. We also assessed the levels of p-S6 and p-4E-BP1 in *Pgam1* KO CD8 T cells at 8 h and 24 h after TCR stimulation (**Fig. 5a**). The phosphorylation levels of both proteins in *Pgam1* KO CD8 T cells at 8 h were comparable to those in wild-type CD8 T cells. In contrast, the phosphorylation levels of both proteins were lower in *Pgam1* KO CD8 T cells at 24 h. These results indicate that the dysfunction of the mTORC1 signal is correlated with the reduction of the glycolytic defect in *Pgam1* KO CD8 T cells.

3. The topic is about T cell immune response, and the deficiency of Pgam1 influenced the metabolism and proliferation of CD8 T cell. Since the CD8+ T cell is crucial for antitumor immunity, what is the role of Pgam1 in T cell-mediated tumor immunity? Or if there are some experiments about the influence of Pgam1 on CD8 T cell effector function?

Response: We agree with reviewer's comment that it is important to assess the role of *Pgam1* in T-cell mediated tumor immunity. However, it will take a long period to determine the anti-tumor activity of CD8 T cells. We have only obtained the preliminary results on the anti-tumor response in T cell-specific *Pgam1* KO mice. Therefore, the results are shown in the **Figure for the reviewers**.

4. The authors described the influence of Pgam1 on CD4 and CD8 T cell has some differences, but what is the mechanism of this?

Response: As the reviewer suggested, *Pgam1* deficiency significantly reduced effector differentiation in CD8 T cells. In CD4 T cells, the development of IL-4-producing Th2 cells and Th17 cells was severely impaired, whereas Treg and Th1 differentiation was

less affected. The different Pgam1-dependent metabolic dependency for the differentiation of effector CD8 T cells and of each of the helper T cell subsets may cause these differences, as we mentioned in the Discussion section.

5. According to paper's description, the level of glutamine is lower in Pgam1 KO activated CD8 T cell than in wild-type activated CD8 T cell (Fig. 6a), there is no significant difference between WT and KO. And the experiments after are all about glutamine. So if there are more evidence showed the relation between Pgam1 and glutamine?

Response: We performed metabolic profiling in *Pgam1* CD8 T cells at 6 h and 24 h after TCR stimulation. The data indicated the reduction of the intracellular concentrations of Gln and Glu in *Pgam1* KO CD8 T cells at 24 h, but not at 6 h, after TCR stimulation. The results are shown in **Fig 6a**.

Minor comments:

1. There is a minor mistake in figure legend of Supplementary Fig.1. There are two SFig.1d without Fig.1e which is not consistent with paper.

Response: Thank you for pointing this out. We have corrected these mistakes.

2. In Fig.1d, there is significant difference of ATP between WT and KO in 6h. Why the paper described that the ATP in 6h is comparable?

Response: Since the difference in the ATP concentration between WT and *Pgam1* KO CD8 T cells at 6 h was very small, we verified the ATP concentration. The ATP level of *Pgam1* KO CD8 T cells at 6 h after TCR stimulation was comparable to that of WT cells (**Fig. 1d**).

3. In the result of Impaired TH1, TH2 and TH17 cell differentiation in Pgam1 KO naïve CD4 T cells, according to the paper's description, the expression of CD40L (Supplementary Fig. S4a) is decreased, which is contrast to the results showed in Supplementary Fig. S4a, which showed marked increase.

Response: We have corrected this mistake. The bold line indicates *Pgam1* KO CD4 T cells, whereas the thin line shows wild-type CD4 T cells.

4. For the phosphorylation of S6 and 4E-BP1, whether the stimulation for 36h is too long to test the phosphorylation status?

Response: According to reviewer's suggestion, we measured the phosphorylation of S6 and 4E-BP at 6 h and 24 h (**Fig. 5a** and **Fig. 6e**).

Reviewer #2

(Remarks to the Author):

Toriyama et al., present a mouse model of T-cell specific glycolytic impairment by T cell specific Pgam1 deletion. The authors show compelling in vitro and in vivo data from multiple models of functional T cell defects in the CD4 and CD8 subsets. By tracking antigen specific responses in vivo, the authors demonstrate that the Pgam1-deficient T cells fail to accumulate in infected animals, and show functional perturbations. The authors provide evidence to show that both the CD8 and CD4-Th17 responses are strongly inhibited, leading to 1) failure to clear infection efficiently, and 2) protection against induction of IBD, respectively. To provide mechanistic insight, the authors performed basic metabolic and signaling characterization of the Pgam1-deficient and WT T cells in vitro and conclude that mTORC1 signaling, glycolysis and glutamine form an amplification loop to induce proliferation, differentiation and T cell-dependent immune responses.

Where the cellular and in vivo data of the mouse model as shown in figure 1-4 are compelling and novel, the link from this part of the manuscript to the following 3 figures is somewhat abrupt, leading into mechanisms that were not clearly linked back to the role of Pgam in T cells. Further elucidation of the role of Pgam and the metabolites handled by this enzyme in the T cell subsets identified in the in vivo models would greatly strengthen the manuscript (even from in vitro differentiated T cells). In addition, this reviewer believes that the manuscript as presented falls short to substantiate the

mechanistic claims regarding the cross talk or amplification loop between mTORC1 signaling, glucose and glutamine metabolism and T cell function, and misses some references to relevant publications related to this cross talk.

Appreciation to Reviewer #2

We appreciate this reviewer's helpful comments, which enabled us to improve the quality of our manuscript. Based on the reviewer's suggestions, we have performed several new experiments and discussed the crosstalk between the expression of c-Myc, mTOR, glycolysis and glutamine availability. In addition, we added several references that are important for the discussion in our manuscript.

Concerns:

1. The authors show compelling evidence that ECAR and OCR are affected by Pgam1 deletion, but these assays were performed at 36 hours after activation, right around the time that T cells start proliferating and asymmetrically inherit cMyc (Verbist et al., Nature. 2016 Apr 21; 532(7599): 389–393. not cited by the authors). Low cMyc cells do not develop into full effectors, display low CD25 expression (could explain lower STAT3 signaling), mTOR signaling, fail to sustain numbers in vivo, and do not undergo metabolic remodeling. In a different study it was shown that cMyc protein expression is sensitive to Slc1a5 inhibition (Glutamine transporter) in NK cells (Loftus et al., Nat Commun. 2018 Jun 14;9(1):234). This suggests that there is intricate two-way cross talk between the expression of cMyc, mTOR and glutamine availability in immune cells, and should be discussed, especially in light of the conclusions drawn.

Response: According to the reviewer's comments, we carefully discussed the crosstalk between the expression of c-Myc, mTOR, glycolysis and glutamine availability in activated CD8 T cells in the DISCUSSION section. In addition, we included references that demonstrate the role of c-Myc in lymphocytes.

2. The authors show that proximal signaling is intact during the first 2-6 hours of cells (figure 5), but the obvious changes happen at the moment that the cell cycle would be engaged at 36 hours. In light of this, how do previous findings as described under point 2 link to the current study? Is the perturbation of the cell cycle here the cause or the

effect?

Response: We demonstrated that the phosphorylation status of S6 and 4E-BP at 6 h and 24 h were reduced by *Pgam1* deficiency (**Fig. 5a**) and glutamine deprivation (**Fig. 6e**). At 24 h after stimulation, the point which before first division, the TCR-signaling was attenuated and the c-Myc expression was reduced (**Fig. 1e** and **Fig. 5b**). Furthermore, *Pgam1* KO CD8 T cells showed limited division after TCR-stimulation (**Fig. 2c**). These results suggest that the impairment of signal transduction was caused by glycolytic failure and the reduced intracellular level of glutamine, but not by the asymmetric division of c-Myc. Thus, we conclude that glycolysis, mTORC and glutamine form a feedforward loop.

3. The use of Stat3 as a loading control for western blots seems to be an odd choice in differentially activated cells. Additionally, the loading based on Stat3 is obviously not equal in the blots shown in figures 6e, 7b, 7c, 7d and 7e, making the differences as shown hard to interpret. A more conventional loading control should be included that reflects either cell numbers lysed or total protein on the membrane.

Response: According to the reviewer's suggestion, we used β -actin as a loading control for Western blotting. The results are shown in **Figs. 1e, 1f, 5b, 6f, 7b, 7c, 7d, 7e** and **Supplementary Fig.S1c**.

*4. The metabolic data as presented in figures 3-6 are lacking in addressing the complexity of the pathways impacted by *Pgam1* deletion, and are focused on glutamine depletion, and treatment with the mTORC1 inhibitor rapamycin. The title of figure 6 states 'glycolysis-dependent glutamine import' but the data in figure 6a indicate no significant differences in glutamine, making the link to the figure title and glutamine depletion somewhat odd. The lower pools of glutamate in the KO cells could be the result of altered activity of GLS (glutamine to glutamate), altered assimilation of glutamate in the TCA cycle (via 2-OG), or the increased exchange of glutamate for the import of cystine (by xCT11) to sustain glutathione synthesis.*

Response: We performed metabolic profiling of *Pgam1* KO CD8 T cells at 6 h and 24 h after TCR stimulation and demonstrated that the intracellular concentrations of glutamine and glutamate in *Pgam1* KO CD8 T cells were significantly reduced at 24 h, but not at 6 h, after TCR stimulation (**Fig 6a**). Moreover, we confirmed that the reduction of the intracellular glutamine and glutamate levels at 24 h after TCR stimulation was also induced by 2-DG treatment (**Fig. 6b**). These results indicate that the increased uptake of glutamine after TCR stimulation is dependent on glycolysis in CD8 T cells.

We did not detect intracellular 2-OG in our experimental setting. However, the intracellular concentrations of succinate, fumarate was decreased, whereas the levels of citrate and cis-aconitates remained unchanged at 24 h after TCR stimulation (**Supplementary Fig. S4c**). It seems unlikely that the increased assimilation of Glu in the TCA cycle via 2-OG caused the reduction of the intracellular Glu concentration in *Pgam1* KO CD8 T cells.

Unfortunately, we did not detect Cys by metabolic profiling in our experimental setting. However, we confirmed that the expression of *Slc7a11* was decreased in activated CD8 T cells by the *Pgam1* deficiency and the depletion of extracellular Gln (Our unpublished observation). Thus, it seems unlikely that the increased *Slc7a11* activity induced the reduction of intracellular Glu in *Pgam1* KO activated CD8 T cells.

5. The authors highlight the function of Pgam1 in the introduction on page 4 as a mediator of glycolysis, the PPP and Serine synthesis. However, the 3-PG accumulation can inhibit the PPP in generating nucleotides (important for proliferation), serine was previously shown to be important for T cell proliferation and the generation of one-carbon units for nucleotide synthesis (Ma et al., Cell Metab. 2017 Feb 7;25(2):345-357),

As glutamine is a biosynthetic precursor and important nitrogen source important for nucleotide synthesis, depletion might overlap with the Pgam1 KO phenotype, but might not be the mechanistic equivalent. An informative assay to address this further could be the supplementation of Pgam1 KO T cells with cell permeable metabolic intermediates (such as dimethyl-modified TCA intermediates), or alternative 3-carbon substrates that bypass the block at 3-PG, such as alanine, pyruvate or lactate.

Response: The intracellular concentration of intermediates of the pentose phosphate pathway (PPP) at 24 h was moderately increased in *Pgam1* KO CD8 T cells in comparison to wild-type cells (**Supplementary Fig. S4d**). In contrast, the intracellular levels of IMP, AMP, GMP and UMP were greatly decreased by *Pgam1* deficiency (**Supplementary Fig. S4e**). These results suggest that *de novo* nucleotide synthesis, but not PPP, is inhibited by *Pgam1* deficiency in activated CD8 T cells. We also demonstrated the reduction of *de novo* nucleotide synthesis by glutamine deprivation (**Supplementary Fig. S13d**).

The serine synthesis seems to be unaffected by *Pgam1* deficiency, since the intracellular concentration of Ser and Gly in *Pgam1* KO CD8 T cells, at both 6 h and 24 h after TCR stimulation was comparable to that in wild-type CD8 T cells (**Supplementary Fig. S12**).

The supplementation of pyruvate failed to restore the development of effector CD8 T cells (**Supplementary Fig. S6**), Th2 (**Fig 3d**) and Th17 cells (**Fig 3f**). In sharp contrast, the production of IFN- γ and TNF- α production was restored by the supplementation of pyruvate (**Fig 3b**). Interestingly, the production of IL-2 was suppressed by pyruvate supplementation in both wild-type and *Pgam1* KO Th1 cells (**Fig 3b**). Furthermore, the supplementation of dimethyl-2-OG failed to restore the effector differentiation and the proliferation of *Pgam1* KO T cells (K.T., M.K. and M.Y. unpublished observation).

Thus, *Pgam1* regulates T cell proliferation and differentiation through several distinct pathways.

Reviewers' comments:

Reviewer #1 (Remarks to the Author):

The authors have provided enough responses to my previous critiques. The revised manuscript is substantially improved.

Reviewer #2 (Remarks to the Author):

Toriyama et al., discuss the importance of Pgam1 In T cell activation. The authors show compelling evidence that loss of Pgam1 leads to a lack of T cell proliferation and differentiation.

In the rebuttal letter and revised manuscript the authors provide additional data regarding the metabolic phenotype resulting from Pgam1 deletion in T cells.

In response to my first concern, the authors now provide additional discussion regarding mTOR, glycolysis and glutamine availability, but this is merely a summary of the observations from a previous studies.

In response to point 2 and 3 of my original comments, the authors now provide additional time points and loading controls for their western blot analysis of signaling. It would be strongly recommended that the authors quantify these blots, and provide the number of experiments that are used to draw these conclusions. An example is the under loading in figure 5b in the KO lane, where this does likely not lead to the amount of loss in signal for the other targets, quantification would help. One concern regarding the data in figure 5a is that the authors mention a reduction in the phosphorylation of S6 and 4E-BP1 at 8h in the rebuttal letter, which I do not think is obvious from the data presented. The authors also state in their revised manuscript on page 8: "The ECAR in Pgam1 KO activated CD8 T cells at 8 h was comparable to that in wild-type CD8 T cells (Supplementary Fig. S4a)", the same was mentioned for OCR (although the lack of response to FCCP in figure 1b should be discussed too) . So the response to my second concern is not in line with the data presented. Regarding their conclusions, I do not think these data justify the conclusion that glycolytic failure precedes an issue with signal transduction, but could still be coinciding.

My concerns in point 3 remain with the loading controls as highlighted in my comments above, where quantification would help. The authors could better align the westerns in figure 1e and 7b, where the data presented makes one wonder what the lanes shown outside of the annotated lanes contains. This looks odd and should be straightened.

I commend the addition of additional experiments with metabolic profiling in response to my fourth comment, but there are some important issues to be considered in the conclusions linked to these experiments. Mentioning the depletion of extracellular glutamine (data not shown) is in this case hard to justify. These data need to be included given the importance of glutamine in the mechanism the authors suggest, as this would indicate that the glutamine is being depleted, which is confusing. Do these cells acquire glutamine from the media or do they not? In addition, the fact that succinate and fumarate pools drop, but not those of citrate could be a result of many factors. This explanation as provided is simplistic at best. The reductive assimilation of glutamine could result in these carbons being incorporated in citrate, allowing continued biosynthesis from glutamine in absence of glycolytic flux. This explanation could be substantiated if the authors provide the extracellular glutamine concentrations over time, something they hint at, but do not show.

Some other concerns are in relation to strong statements like de-novo synthesis of nucleotides in the

absence of stable isotope tracing. This should be nuanced in the text to not overstate non-substantiated phenotypes derived from pool size measurements. The same goes for the last sentence in the rebuttal letter, the authors mention the complexity of the system and likelihood of the loss of Pgam influencing T cell biology, this should also come through in the manuscript. I would also recommend the authors include the in vivo data in the manuscript, now that the survival curves will have been extended and the data will be much clearer one would assume.

In conclusion, I am not sure the authors substantiated the feed forward loop as suggested, irrespective of the addition of additional experiments. I believe that these conclusions detract from the first half of the paper.

RESPONSES TO THE REVIEWERS

Reviewer #1

(Remarks to the Author):

The authors have provided enough responses to my previous critiques. The revised manuscript is substantially improved.

Appreciation to Reviewer #1

We appreciate this reviewer's critical reading and judgment.

Reviewer #2

(Remarks to the Author):

Appreciation to Reviewer #2

We appreciate this reviewer's helpful comments, which enabled us to improve the quality of our manuscript. Based on the reviewer's suggestions, we have performed several new experiments and revised the DISCUSSION section.

Toriyama et al., discuss the importance of Pgam1 In T cell activation. The authors show compelling evidence that loss of Pgam1 leads to a lack of T cell proliferation and differentiation.

In the rebuttal letter and revised manuscript the authors provide additional data regarding the metabolic phenotype resulting from Pgam1 deletion in T cells.

In response to my first concern, the authors now provide additional discussion regarding mTOR, glycolysis and glutamine availability, but this is merely a summary of the observations from a previous studies.

In response to point 2 and 3 of my original comments, the authors now provide additional time points and loading controls for their western blot analysis of signaling. It would be strongly recommended that the authors quantify these blots, and provide the number of experiments that are used to draw these conclusions. An example is the

under loading in figure 5b in the KO lane, where this does likely not lead to the amount of loss in signal for the other targets, quantification would help.

Response: According to the reviewer's suggestion, we quantified the intensity of the blots using the Image J software program (<http://rsb.info.nih.gov/ij>). The relative intensity is indicated in the figures. The number of independent experiments that we performed is also included in the figure legends.

One concern regarding the data in figure 5a is that the authors mention a reduction in the phosphorylation of S6 and 4E-BP1 at 8h in the rebuttal letter, which I do not think is obvious from the data presented. The authors also state in their revised manuscript on page 8: "The ECAR in Pgam1 KO activated CD8 T cells at 8 h was comparable to that in wild-type CD8 T cells (Supplementary Fig. S4a)", the same was mentioned for OCR (although the lack of response to FCCP in figure 1b should be discussed too) . So the response to my second concern is not in line with the data presented. Regarding their conclusions, I do not think these data justify the conclusion that glycolytic failure precedes an issue with signal transduction, but could still be coinciding.

Response: We mentioned that the phosphorylation levels of p-S6 and p-4E-BP1 in Pgam1 KO CD8 T cells at 8 h were comparable to those in wild-type CD8 T cells in the previous rebuttal letter. We concluded that the phosphorylation levels of both proteins in Pgam1 KO CD8 T cells at 8 h were comparable to those in wild-type CD8 T cells, whereas the phosphorylation levels of both proteins were lower in Pgam1 KO CD8 T cells at 24 h.

According to the to the reviewer's comments, we discussed the lack of responsiveness to FCCP in Pgam1 KO CD8 T cells in the DISCUSSION section. We showed the decreased levels of both NAD⁺ and NADH in Pgam1 KO CD8 T cells at 24 h after TCR stimulation (**Supplementary Fig. S4d**). We also assessed the responsiveness of Pgam1 KO CD8 T cells to FCCP without oligomycin treatment and confirmed the lack of response to FCCP (**Fig. for reviewers**).

In addition, the conclusion that glycolytic failure precedes an issue with signal transduction was modified. We concluded that both glycolysis and glutamine may be required to activate signal transduction in activated T cells via the supplementation of metabolic products, such as ATP. We mentioned this point in the DISCUSSION section.

My concerns in point 3 remain with the loading controls as highlighted in my comments above, where quantification would help. The authors could better align the westerns in figure 1e and 7b, where the data presented makes one wonder what the lanes shown outside of the annotated lanes contains. This looks odd and should be straightened.

Response: Although it was requested that the of western blotting be straightened (**Fig. 1e and 7b**), this was difficult to perform due to technical limitations. The outside of the annotated lanes contains lanes under different conditions that were not required to draw the conclusion of our current manuscript.

I commend the addition of additional experiments with metabolic profiling in response to my fourth comment, but there are some important issues to be considered in the conclusions linked to these experiments. Mentioning the depletion of extracellular glutamine (data not shown) is in this case hard to justify. These data need to be included given the importance of glutamine in the mechanism the authors suggest, as this would indicate that the glutamine is being depleted, which is confusing. Do these cells acquire glutamine from the media or do they not? In addition, the fact that succinate and fumarate pools drop, but not those of citrate could be a result of many factors. This explanation as provided is simplistic at best. The reductive assimilation of glutamine could result in these carbons being incorporated in citrate, allowing continued biosynthesis from glutamine in absence of glycolytic flux. This explanation could be substantiated if the authors provide the extracellular glutamine concentrations over time, something they hint at, but do not show.

Response: According to the reviewer's suggestion, we measured the extracellular glutamine concentrations in wild-type and *Pgam1* KO CD8 T cells at 6 h and 24 h after

TCR stimulation. These results are included in the revised manuscript (**Supplementary Fig. 12**). The extracellular concentration of glutamine in *Pgam1* KO CD8 T cell cultures was comparable to that in wild-type CD8 T cells at 6 h after TCR stimulation. In contrast, the level of extracellular glutamine in *Pgam1* KO CD8 T cell cultures was significantly higher than that in wild-type CD8 T cells at 24 h. These data support the reduction of glutamine uptake in *Pgam1* KO CD8 T cells at 24 h after TCR stimulation. The intracellular concentrations of glycolytic intermediates (F1-6P, DHAP, glycerophosphate, 3PG and lactate) under glutamine-deprived conditions were also included in **Supplementary Fig. 14b**.

As the reviewer pointed out, the intracellular concentration of citrate in *Pgam* KO CD8 T cells was comparable to that in wild-type CD8 T cells, whereas the levels of succinate, fumarate and malate were decreased. The citrate was synthesized through several different sources, including glucose and glutamine. We showed that the Srebp1 protein level was decreased in the nuclei of *Pgam1* KO CD8 T cells at 24 h after TCR stimulation (**Fig. 5b**). In addition, we included the data showing the reduced expression of acetyl-CoA citrate lyase (*acly*) mRNA in *Pgam1* KO CD8 T cells at 24 h and 48 h after TCR stimulation (**Fig. 5e**). These data suggest that lipid and sterol synthesis was decreased in *Pgam1* KO activated CD8 T cells. Therefore, we consider that the intracellular concentration of citrate was not decreased in *Pgam1* KO CD8 T cells, despite the decreased glycolysis and glutamine uptake, since the lipid and sterol synthesis was impaired.

Some other concerns are in relation to strong statements like de-novo synthesis of nucleotides in the absence of stable isotope tracing. This should be nuanced in the text to not overstate non-substantiated phenotypes derived from pool size measurements. The same goes for the last sentence in the rebuttal letter, the authors mention the complexity of the system and likelihood of the loss of Pgam influencing T cell biology, this should also come through in the manuscript. I would also recommend the authors include the in vivo data in the manuscript, now that the survival curves will have been extended and the data will be much clearer one would assume.

In conclusion, I am not sure the authors substantiated the feed forward loop as

suggested, irrespective of the addition of additional experiments. I believe that these conclusions detract from the first half of the paper.

Response: As the reviewer pointed out, it is difficult to assess the status of *de novo* nucleotides synthesis without stable isotope tracing. We changed “de novo nucleotides synthesis” to “intracellular nucleotide concentration”.

Although the results of anti-tumor activity in T-cell specific *Pgam1* KO mice are potentially important, a further analysis of the molecular and cellular mechanisms would be required to reach a conclusion. Furthermore, the results regarding the anti-tumor activity are not essential to reach the conclusion of our current manuscript. Thus, we excluded the anti-tumor activity data from our current manuscript. We are planning include these results regarding the anti-tumor activity in T-cell specific *Pgam1* KO mice in another report.

As the reviewer pointed out, the metabolic pathways in activated T cells are highly complicated. Thus, we have withdrawn our conclusion that mTORC1, glycolysis and glutamine form a feedforward loop to induce proliferation, differentiation, and T cell-dependent immune responses. In the revised manuscript, we conclude that mTORC1, glycolysis and glutamine affect each other and cooperate to induce T cell proliferation and differentiation. Thus, the loss of *Pgam1* influences the T cell-dependent immune response through multiple pathways.

REVIEWERS' COMMENTS:

Reviewer #2 (Remarks to the Author):

The reviewers have provided additional revisions in the text that nuance the conclusions. These revisions and additions to the text and discussion have addressed my previous concerns.

RESPONSES TO THE REVIEWERS

Reviewer #2

(Remarks to the Author):

The reviewers have provided additional revisions in the text that nuance the conclusions. These revisions and additions to the text and discussion have addressed my previous concerns.

Appreciation to Reviewer #2

We appreciate this reviewer's critical reading and judgment.